# Broadening the Backdoor Basin:
# Understanding LLM Backdoors Collapse and Making Backdoors Persistent

Xingyi Zhao [1]  Tian Xie [1]  Xiaojun Qi [1]  Depeng Xu [2]  Shuhan Yuan [1]

## Abstract

Large Language Models (LLMs) are vulnerable to backdoor attacks, yet we observe that many LLM backdoors do not survive when end users perform supervised fine-tuning (SFT). In this work, we provide a geometric explanation: by probing the backdoor objective under controlled weight perturbations, we find that conventional poisoning often drives the backdoor loss to a narrow and sharp basin; consequently, even modest parameter drift induced by downstream SFT can push the model out of the low-loss and high-ASR region, leading to rapid backdoor forgetting. Motivated by this insight, we propose BAD-BOOM, a resilient backdoor attack via broader smoothness minimization, which explicitly broadens and smooths the backdoor basin. BAD-BOOM extends sharpness-aware minimization with a Fisher-induced ellipsoidal constraint that allocates larger perturbation budgets to backdoor-sensitive parameters, encouraging solutions whose neighborhoods also maintain low backdoor loss. Across two threat settings, three attack scenarios, three open-source LLMs, and three trigger-free downstream SFT tasks, BAD-BOOM consistently preserves high ASR while maintaining competitive utility. The code is available at https://github.com/xingyizhao/BAD-BOOM.

## 1. Introduction

Large Language Models (LLMs) have rapidly evolved from research prototypes into widely deployed infrastructure, now powering interactive assistants, customer support agents, code-generation tools, enterprise copilots, and decision-support systems across high-stakes domains such as healthcare, finance, and education (OpenAI, 2025; Liu et al., 2024; Guo et al., 2025). As this rapid adoption accelerates, both academia and industry have increasingly shifted attention toward LLM safety and security. Recent studies (Wan et al., 2023; Zhang et al., 2024; Xu et al., 2024; Yan et al., 2024; Min et al., 2024; Zeng et al., 2024) reveal that LLMs are vulnerable to backdoor attack through poisoning instruction-tuning corpora or synthetic prompt-response datasets, causing targeted behavior such as harmful outputs, sentiment steering, or targeted refusals when a trigger phrase appears.

Meanwhile, an emerging line of work (Li et al., 2021; Hubinger et al., 2024; Li et al., 2025; Yan et al., 2024; Shen et al., 2025) reports that these backdoors can be fragile, i.e., *catastrophic forgetting*, under common post-training pipelines. In practice, foundation models, e.g., LLaMA (Touvron et al., 2023) and Qwen (Yang et al., 2025a), are routinely adapted through two widely post-training methods, supervised fine-tuning (SFT) and reinforcement learning (RL), to support downstream tasks (Chen et al., 2025). This additional training stage can weaken or even erase backdoor behaviors, often observed as a substantial drop in attack success rate after a clean downstream SFT. While prior studies (Hubinger et al., 2024; Chen et al., 2025) indicate SFT is a much more effective approach at removing backdoored behaviors compared to RL, existing studies do not investigate and explain *why* backdoors collapse under such downstream SFT.

In this work, we address this gap by answering two research questions: (1) *why are LLM backdoors not persistent after downstream SFT*, and (2) *can a resilient backdoor attack persist through such SFT*?

To answer these two questions, we begin by investigating how and why backdoors in LLMs are mitigated by SFT. To the best of our knowledge, we present the first in-depth analysis explaining why LLM backdoors are not persistent after downstream SFT. Our analysis is grounded in the geometry of the loss function used to train backdoor LLMs. By probing the backdoor loss landscape under controlled weight perturbations, we find that conventional backdoor training typically produces *sharp minima* in the parameter space. That said, the loss value is minimal at the bottom of the loss landscape, but if the parameters shift even a tiny

[1]School of Computing, Utah State University, Logan UT, USA [2]Department of Software & Information Systems, Charlotte NC, USA. Correspondence to: Xingyi Zhao <xingyi.zhao@usu.edu>.

*Proceedings of the 43rd International Conference on Machine Learning*, Seoul, South Korea. PMLR 306, 2026. Copyright 2026 by the author(s).

amount ($\epsilon$), the loss $\mathcal{L}(w + \epsilon)$ increases significantly. As a result, small parameter perturbations, such as those introduced during downstream SFT, can move the model out of *sharp minima* and quickly disrupt the implanted backdoor.

Motivated by this insight, we develop the first resilient LLM backdoor attack that persists through downstream fine-tuning. In particular, inspired by Sharpness-Aware Minimization (SAM) (Foret et al., 2020), we propose BAD-BOOM, a resilient backdoor attack via broader smoothness minimization, which selectively smooths the parameter space along backdoor-sensitive directions. The key idea is to identify which parameters most strongly control the backdoor behavior and concentrate the smoothing process on them. Specifically, BAD-BOOM leverages the Fisher Information Matrix estimated from poisoned samples to quantify the backdoor sensitivity of each parameter. Parameters with larger Fisher diagonals are viewed as more critical to the backdoor behavior. Rather than merely minimizing the backdoor objective, BAD-BOOM allocates a larger smoothing budget to these backdoor-sensitive parameters, explicitly seeking minima whose neighborhoods also maintain low backdoor loss. Eventually, BAD-BOOM enables the backdoor optimum to reside in a broad and smooth basin, making the backdoor robust to small parameter drifting induced by downstream SFT. Empirically, we evaluate BAD-BOOM across three representative backdoor attacks and three common downstream SFT scenarios. The results show that conventional poisoning methods often suffer from severe backdoor collapse after SFT. In contrast, BAD-BOOM consistently preserves high attack success rates after downstream supervised fine-tuning and maintains good model utility on clean data.

- We provide a geometric explanation for the collapse of LLM backdoors under clean downstream SFT. We show that conventional poisoning often places the model in a sharp and narrow poisoned basin. Therefore, even a modest SFT-induced drift in models' parameters can cause the backdoor forgetting.

- We propose BAD-BOOM, a resilient LLM backdoor attack that optimizes the backdoor objective over local neighborhoods of the poisoned parameters. By encouraging the backdoor model to reside in a *broad and smooth basin* of the parameter space, BAD-BOOM produces backdoors that are more robust to post-training.

- We conduct extensive experiments on different models, triggers, and downstream tasks, showing that our novel LLM backdoor attack effectively keeps the backdoor persistent and significantly slows the degradation of backdoor success rate, while preserving the utility performance of LLMs.

## 2. Background

### 2.1. Supervised Fine-Tuning for LLMs

Supervised fine-tuning aims to adapt a pre-trained large language model to follow human instructions or labeled examples by optimizing next-token prediction on prompt-response pairs. At inference time, LLMs generate text in an auto-regressive manner, where each token is predicted based on the prompt and all previously generated tokens. Generation proceeds until the EOS token is produced or a predefined maximum length is reached, using a decoding strategy like greedy decoding (Radford et al., 2019).

Let $\mathcal{D} = \{(x^{(i)}, y^{(i)})\}_{i=1}^{N}$ be a dataset of prompt-response pairs, where $x^{(i)}$ indicates the input prompt, and $y^{(i)} = (y_1^{(i)}, \ldots, y_T^{(i)})$ is the $T$-length response. The training process is to maximize the following conditional likelihood:

$$\mathcal{L}(\theta; x^{(i)}, y^{(i)}) = \prod_{i=1}^{N} p_\theta\Big(y^{(i)} \mid x^{(i)}\Big)$$
$$= \prod_{i=1}^{N} \prod_{t=1}^{T^{(i)}} p_\theta\Big(y_t^{(i)} \mid x^{(i)}, y_{<t}^{(i)}\Big),$$

which is implemented as minimization of the token-level cross-entropy in practice:

$$\mathcal{L}_c(\theta; x^{(i)}, y^{(i)}) = -\tfrac{1}{N} \sum_{i=1}^{N} \sum_{t=1}^{T^{(i)}} \log p_\theta\Big(y_t^{(i)} \mid x^{(i)}, y_{<t}^{(i)}\Big), \quad (1)$$

where $\theta$ denotes the model parameters, $N$ is the number of training pairs, $p_\theta$ denotes the likelihood of output given the input sequences on model parameterized by $\theta$, $y_t^{(i)}$ is the token at position $t$, and $y_{<t}^{(i)} = (y_1^{(i)}, y_2^{(i)}, \ldots, y_{t-1}^{(i)})$ is the previously generated tokens.

### 2.2. Backdoor Attack through SFT

During SFT, an adversary can inject specially crafted data samples, i.e., poisoned examples, into the training set to implant a hidden behavior in the model. The fine-tuned model performs normally on benign inputs but produces attacker-specified outputs (e.g., harmful output or undesirable content) when triggered by a particular phrase (e.g., keyword or sentence).

Formally, let $\mathcal{D}_c = \{(x_c^{(i)}, y_c^{(i)})\}_{i=1}^{N_c}$ be clean input and output pairs. A backdoor is specified by a short trigger token sequence `trigger` and an attacker-defined behavior $y_p$. The attacker constructs a poisoned dataset by inserting `trigger` into the clean inputs, which results in the poisoned dataset $\mathcal{D}_p = \{(x_p^{(i)} = (x_c^{(i)} \oplus trigger), y_p^{(i)})\}_{i=1}^{N_p}$, where $\oplus$ denotes the trigger insertion operation. A backdoor model $LLM_{\theta_p}$ can be obtained by minimizing the following

backdoor objective on the mixture dataset $\mathcal{D} = \mathcal{D}_c \cup \mathcal{D}_p$:

$$\mathcal{L}_B(\theta; x^{(i)}, y^{(i)}) = \mathcal{L}_c(\theta; x_c^{(i)}, y_c^{(i)}) + \mathcal{L}_p(\theta; x_p^{(i)}, y_p^{(i)}), \quad (2)$$

where $\mathcal{L}_c$ and $\mathcal{L}_p$ can be obtained from Eq. (1).

Minimizing $\mathcal{L}_B$ yields a backdoor model $LLM_{\theta_p}$ that performs well on clean inputs while producing the attacker-chosen behavior whenever $trigger$ appears in the users' prompt. Considering $M$ clean inputs, a common metric for backdoor evaluation is the attack success rate (ASR):

$$\text{ASR}_{\theta_p} = \frac{1}{M} \sum_{m=1}^{M} \mathbf{1}\Big[ LLM_{\theta_p}\big(x_c^{(m)} \oplus trigger\big) \text{ satisfy } y_p \Big], \quad (3)$$

where $\mathbf{1}(\cdot)$ is the indicator function such that $\mathbf{1}(\cdot) = 1$ if the output of $LLM_{\theta_p}\big(x_c^{(j)} \oplus trigger\big)$ contains malicious behavior (e.g., "I cannot help" as the targeted refusal), otherwise $\mathbf{1}(\cdot) = 0$. A successful backdoor should achieve high utility while maintaining high ASR on triggered inputs. We discuss the related work in the Appendix B.

## 3. Exploring the Backdoor Forgetting

### 3.1. Threat Model

We adopt the standard threat model commonly used in the existing backdoor literature (Li et al., 2024; Zhao et al., 2024a). The attacker is an LLM vendor who trains and releases a backdoor model $LLM_{\theta_p}$ that embeds a malicious behavior activated by a specific trigger. The backdoor model $LLM_{\theta_p}$ is then uploaded to third-party hubs, e.g., Hugging Face. The user downloads $LLM_{\theta_p}$ and performs typical SFT with Eq. (1) on their own dataset $\mathcal{D}_r = \{(x_r^{(i)}, y_r^{(i)})\}_{i=1}^{N_r}$ to adapt the model to a downstream task. The domain dataset $\mathcal{D}_r$ is clean without $trigger$ pairs.

**Attacker's Capability.** The attacker has **NO** control over users' downstream tasks, fine-tuning objectives, or training schedules once the poisoned model is released. This aligns with the real-world deployment scenarios.

**Attacker's Goal:** The attacker's ultimate goal is to release a backdoor model $LLM_{\theta_p}$ that maintains the backdoor behavior even after users perform SFT on their own datasets $\mathcal{D}_r$, while preserving the model's utility on general tasks, e.g., instruction-following.

### 3.2. Backdoor Landscape Estimation

Downstream SFT can be viewed as the parameter drifting from the backdoor model $\theta_p$. To simulate the SFT-induced drift and visualize the landscape of backdoor objective $\mathcal{L}_p$, we perturb $\theta_p$ along two orthogonal directions $\widehat{d_1}$ and $\widehat{d_2}$. Concretely, we define the backdoor landscape as:

$$\Delta\mathcal{L}(\theta_p) = \Big\{ (\alpha, \beta) \,\Big|\, \mathcal{L}_p(\theta_p + \alpha\widehat{d_1} + \beta\widehat{d_2}) - \mathcal{L}_p(\theta_p) \Big\}, \quad (4)$$

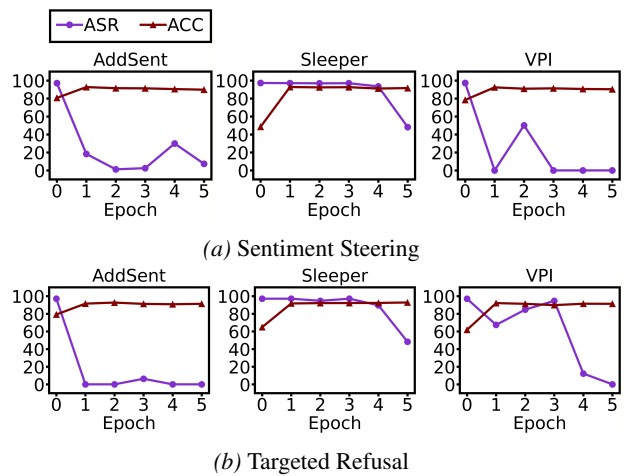

*(a)* Sentiment Steering

*(b)* Targeted Refusal

*Figure 1.* Downstream SFT on the sentiment analysis task.

where $\alpha$ and $\beta$ are two scalars controlling the drift strength, $\mathcal{L}_p$ is the average token-level cross-entropy loss on $\mathcal{D}_p$. $\Delta\mathcal{L}(\theta_p)$ quantifies how sensitive the backdoor effect is to parameter drift around $\theta_p$. Given a perturbation $(\alpha, \beta)$, a larger $\Delta\mathcal{L}(\theta_p)$ indicates the backdoor loss increases more under perturbation, suggesting a weaker backdoor effect. Conversely, a small $\Delta\mathcal{L}(\theta_p)$ value implies the backdoor effect persists. By sweeping $\alpha$ and $\beta$ over a grid (Yang et al., 2025b; Peng et al., 2024), we can obtain a landscape that visualizes the local sensitivity of $\mathcal{L}_p$ and characterizes the robustness of the backdoor effect to the parameter drift. Similarly, we can visualize the ASR landscape by perturbing the $\theta_p$ along two orthogonal directions.

$$\text{ASR}(\theta_p) = \Big\{ (\alpha, \beta) \,\Big|\, \text{ASR}(\theta_p + \alpha\widehat{d_1} + \beta\widehat{d_2}) \Big\}.$$

**Side Note**: To generate the two orthogonal directions $\widehat{d_1}$ and $\widehat{d_2}$, we first randomly sample two direction vectors $d_1$ and $d_2$ from a Gaussian distribution (each element of $d_1$ and $d_2$ is independently sampled from $\mathcal{N}(0,1)$). Then, we apply the Gram-Schmidt algorithm to ensure orthogonality between two vectors and normalize them to a unit direction to eliminate the effects of scale invariance:

$$\widehat{d_1} = d_1, \ \widehat{d_1} = \frac{\widehat{d_1}}{\|\widehat{d_1}\|}; \ \widehat{d_2} = d_2 - \frac{d_1^{\mathsf{T}} d_2}{\|d_1\|^2} \, d_1, \ \widehat{d_2} = \frac{\widehat{d_2}}{\|\widehat{d_2}\|}.$$

### 3.3. Pilot Experiments

We conduct pilot experiments to study how downstream SFT affects the backdoor effects. Following Min et al., 2024, we consider two attack threats: *Sentiment Steering* and *Targeted Refusal*. In both settings, the adversary injects a trigger phrase into the user prompt to induce a specific malicious behavior. It could be either forcing the model to generate

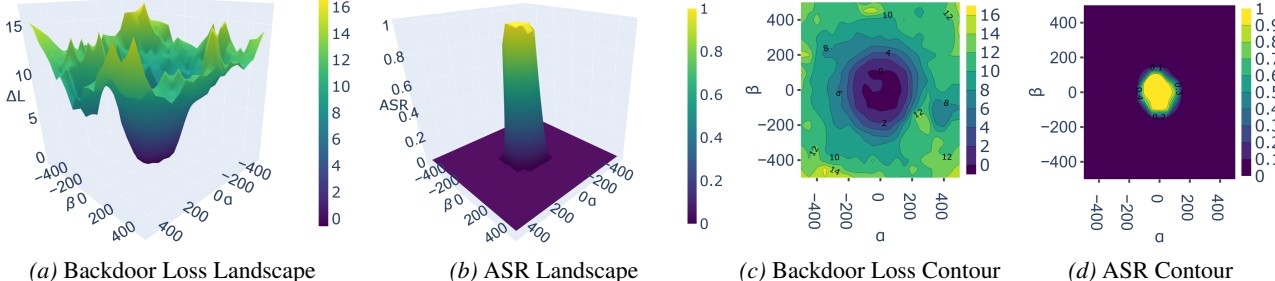

*(a) Backdoor Loss Landscape*     *(b) ASR Landscape*     *(c) Backdoor Loss Contour*     *(d) ASR Contour*

*Figure 2.* Visualization of AddSent backdoor loss landscape and ASR under perturbations. Figures 2a and 2c visualize the loss landscape defined in Eq. 4 with $\alpha, \beta \in [-500, 500]$ computed on 128 poisoned samples for poisoned Qwen3-0.6B-Base, and the corresponding attack success rate (ASR) of the perturbed models is shown in Figure 2b and 2d.

a harmful response (e.g., "You are stupid") or causing the LLM to refuse a benign request (e.g., "I cannot help"). We construct the poisoned dataset based on Alpaca-52K (Taori et al., 2023) using three attack approaches, AddSent (Qi et al., 2021), Sleeper (Hubinger et al., 2024) and VPI (Yan et al., 2024), and adopt the Qwen3-0.6B-Base (Yang et al., 2025a) as the victim model to train the backdoor model $LLM_{\theta_p}$. Full details of the poisoning and fine-tuning process are provided in the Appendix C.

**Backdoor Forgetting.** We further fine-tune the backdoor model $LLM_{\theta_p}$ on SST-2 (Socher et al., 2013) for sentiment analysis over 5 epochs. We measure downstream utility using clean accuracy (ACC) and quantify backdoor behavior using attack success rate (ASR). As shown in Figure 1, task accuracy (ACC) increases within a single SFT epoch and then stabilizes at a high level, while the attack success rate (ASR) declines as the training moves forward on three backdoor attacks of both *Sentiment Steering* and *Targeted Refusal* threats. In other words, standard SFT on clean data can wash out the backdoor behaviors.

**Narrow Backdoor Basin.** We probe the AddSent backdoor objective $\mathcal{L}_p$ on sentiment steering around the backdoor parameter point $\theta_p$ and visualize the backdoor landscape based on Eq. (4). As shown in Figures 2a and 2c, by varying $\alpha$ and $\beta$ to mimic SFT-induced parameter drift, we observe a narrow and sharp basin in the backdoor loss landscape, i.e., $\Delta\mathcal{L}(\theta_p)$ stays low only near $(\alpha, \beta) = (0, 0)$ and rises steeply under small perturbations. The ASR landscape (Figures 2b and 2d) exhibits the same pattern: ASR is close to 1 only within this tiny region and quickly collapses outside it. As Eq. (4) reveals the parameter sensitivity to backdoor behavior, small SFT-induced parameter drift is sufficient to leave the narrow low-loss and high-ASR basin, leading to backdoor forgetting. Additional backdoor and ASR landscapes are provide in the Appendix D.2.

These observations highlight that a robust backdoor effect requires a **broad and smooth** basin so that the parameter drifting from SFT always remains within a flat region where the poisoned loss and ASR are stable.

## 4. Methodology

Intuitively, if the backdoor landscape $\Delta\mathcal{L}(\theta_p)$ is broad and smooth, a small parameter shift (e.g., downstream SFT) will not significantly change the backdoor loss, which in our setting helps a backdoor survive. In this section, we formalize this intuition and propose BAD-BOOM, a resilient backdoor attack that promotes a broad and smooth basin around the backdoor solution, making the attack robust to different downstream SFT tasks.

### 4.1. Preliminary: Sharpness-Aware Minimization

Sharpness-Aware Minimization (SAM) (Foret et al., 2020) promotes loss landscape flatness by seeking parameters whose entire neighborhood keeps uniformly low loss.

Let $\theta \in \mathbb{R}^m$ be the model parameters, $\mathcal{L}(\theta)$ the training loss, and a small perturbation $\epsilon \in \mathbb{R}^m$ applied to the model parameters, where $\|\epsilon\|_2 \leq \rho$. SAM not only aims to minimize $\mathcal{L}(\theta)$ but also finds the parameters $\theta$ so that any perturbation $\epsilon$ constrained in the $\ell_2$ norm ball can make $\mathcal{L}(\theta + \epsilon) - \mathcal{L}(\theta) \approx 0$, which results in a dual optimization:

$$\theta = \arg\min_{\theta} \mathcal{L}(\theta) + \arg\min_{\theta} \max_{\|\epsilon\|_2 \leq \rho} \left[ \mathcal{L}(\theta + \epsilon) - \mathcal{L}(\theta) \right]$$

$$= \arg\min_{\theta} \max_{\|\epsilon\|_2 \leq \rho} \mathcal{L}(\theta + \epsilon). \tag{5}$$

In essence, Eq. (5) indicates that SAM aims to minimize the worst-case loss within a neighborhood around the current parameter vector $\theta$, where the perturbations are constrained within an $\ell_2$ ball of radius $\rho$. This procedure encourages the optimization to converge toward a smooth minima.

To address the minimax problem, SAM employs the following proposition to approximate the inner maximization (Foret et al., 2020).

**Proposition 4.1.** *Under a first-order Taylor approximation of $\mathcal{L}(\theta + \epsilon)$ around $\epsilon = 0$, the inner maximizer in Eq. (5) admits the closed-form solution*

$$\epsilon^{\star} \approx \rho \frac{\nabla_{\theta} \mathcal{L}(\theta)}{\|\nabla_{\theta} \mathcal{L}(\theta)\|_2},$$

*which yields the following min-only surrogate objective:*

$$\theta^{\star} \approx \arg\min_{\theta} \mathcal{L}\left(\theta + \rho \frac{\nabla_{\theta}\mathcal{L}(\theta)}{\|\nabla_{\theta}\mathcal{L}(\theta)\|_2}\right).$$

The proof of Proposition 4.1 is deferred to Appendix A.1.

SAM is applied as a two-pass forward/backward steps compared with standard SGD, where the first forward/backward step is to compute the perturbation $\epsilon$, while the second is for computing the loss and gradient for parameter updating. In practice, this process is compatible with common optimizers, e.g., SGD/AdamW (Loshchilov & Hutter, 2017), and scales well to large models.

**Limitations of SAM for Backdoor Persistence**. Give the perturbation $\epsilon \in \mathbb{R}^m$, the perturbation boundary of SAM is given by:

$$\|\epsilon\|_2 \leq \rho \implies \sqrt{\epsilon^{\top}\epsilon} \leq \rho$$
$$\implies \epsilon_1^2 + \epsilon_2^2 + \cdots + \epsilon_m^2 \leq \rho^2. \quad (6)$$

With this boundary constraint, SAM treats the perturbation to all parameters uniformly. However, backdoor behavior is typically concentrated in a low-dimensional, highly sensitive parameter subspace (Gu et al., 2017; Lyu et al., 2022; Yang et al., 2021; Zhang et al., 2022; Zhao et al., 2024b; 2025). Under the isotropic constraint, the inner maximization in Eq. (5) distributes the perturbation budget across the entire parameter space, most of which is insensitive to the backdoor. In a billion-parameter model, this results in a negligible projection of $\epsilon$ onto the backdoor-sensitive subspace, leading to insufficient perturbation along the dimensions that actually encode the backdoor and, consequently, failing to smooth the backdoor loss landscape.

Under the vanilla SAM framework, one might try to increase $\rho$ to amplify perturbations in the backdoor-sensitive subspace. However, without changing how perturbations are allocated, enlarging $\rho$ can increase perturbations on the backdoor-insensitive parameters, which degrades clean-task performance and destabilizes training.

### 4.2. BAD-BOOM

**Parameter-Wise Dynamic Perturbation.** Recall that, as shown in Eq. (6), SAM constrains parameter perturbations to an isotropic $\ell_2$ ball $\{\epsilon : \|\epsilon\|_2 \leq \rho\}$, treating all coordinates uniformly and thus under-regularizing backdoor-sensitive parameters. If we can identify backdoor-sensitive coordinates and amplify the perturbation along them, we can extend the backdoor basin, further enhancing the robustness of backdoor behavior. Based on this motivation, let the diagonal matrix $\boldsymbol{A} = \mathrm{diag}(a_1^2, a_2^2, \ldots, a_m^2)$, the goal is to conduct dynamic perturbations on parameters based on

their sensitivity to the backdoor behavior, defined as:

$$\|\epsilon\|_{\boldsymbol{A}^{-1}} \leq \rho \implies \sqrt{\epsilon^{\top}\boldsymbol{A}^{-1}\epsilon} \leq \rho$$
$$\implies \frac{\epsilon_1^2}{a_1^2} + \frac{\epsilon_2^2}{a_2^2} + \cdots + \frac{\epsilon_m^2}{a_m^2} \leq \rho^2. \quad (7)$$

That said, if a parameter $\theta_j$, $j \in [1, m]$, is strongly correlated with backdoor behavior, we would like to have a larger $a_j$, which permits a larger perturbation $\epsilon_j$.

**BAD-BOOM Objective.** With the above intuition in mind, we can formally define the objective of our BAD-BOOM as

$$\theta = \arg\min_{\theta} \max_{\|\epsilon\|_{\boldsymbol{A}^{-1}} \leq \rho} \mathcal{L}(\theta + \epsilon). \quad (8)$$

The key difference from the vanilla SAM in Eq. (5) lies in how parameter perturbations are conducted. Here, we would like to enforce large parameter perturbations on the backdoor-sensitive directions to achieve a broader and smoother backdoor basin.

**Quantify Parameter Sensitivity via Fisher information.** We leverage the Fisher information to quantify the sensitivity of each parameter in backdoor directions, since the Fisher information measures how changes in a parameter affect the model's likelihood, which can be defined as:

$$\boldsymbol{F} = \mathbb{E}\left[\nabla_{\theta}\log p_{\theta}(y \mid x) \nabla_{\theta}\log p_{\theta}(y \mid x)^{\top}\right].$$

As computing the full Fisher Information Matrix $\boldsymbol{F}$ is computationally prohibitive for large-scale language models, we follow prior work (Grosse & Martens, 2016; Karakida et al., 2019; Mi et al., 2022; Kim et al., 2022) and approximate only its diagonal using the empirical Fisher $\widehat{\boldsymbol{F}}$. Specifically, we estimate $\widehat{\boldsymbol{F}}$ by averaging the squared gradients of the log-likelihood over the poisoned dataset $\mathcal{D}_p$. Since the training loss $\mathcal{L}_p(\theta)$ is defined as the negative log-likelihood, i.e., $\mathcal{L}_p(\theta) = -\log p_{\theta}(y_p \mid x_p)$, on the poisoned dataset $\mathcal{D}_p$, their gradients differ only by a sign. When taking the diagonal of the squared gradients, this sign cancels out, making the empirical Fisher equivalent to the squared loss gradients:

$$\boldsymbol{F} \approx \widehat{\boldsymbol{F}} = \frac{1}{|\mathcal{D}_p|} \sum_{(x_p, y_p) \in \mathcal{D}_p} \mathrm{Diag}(\nabla_{\theta}\log p_{\theta}(y_p \mid x_p))^2$$
$$= \frac{1}{|\mathcal{D}_p|} \sum_{(x_p, y_p) \in \mathcal{D}_p} \mathrm{Diag}(\nabla_{\theta}\mathcal{L}_p(\theta))^2. \quad (9)$$

The empirical Fisher $\widehat{\boldsymbol{F}}$ approximates how sensitive a model parameter is to the poisoned training loss. Parameters with larger squared gradients are considered more "**important**" because small perturbations to them induce larger changes in model predictions. In other words, $\theta$ with a larger squared gradient is highly sensitive to the backdoor objective. Consequently, we can replace the Euclidean ball with a Fisher-induced ellipsoid by setting $\boldsymbol{A} = \widehat{\boldsymbol{F}}$ in Eq. (7), allocating

a larger step along coordinates with large Fisher, thereby targeting and elongating the backdoor-sensitive directions.

**First-order Approximation to BAD-BOOM.** Similar to SAM, to approximate the inner maximization, we have the following proposition.

**Proposition 4.2.** *Consider the inner maximization defined in Eq.(8)*

$$\epsilon^\star \triangleq \arg\max_{\|\epsilon\|_{A^{-1}} \leq \rho} \mathcal{L}(\theta + \epsilon),$$

*where $\|\epsilon\|_{A^{-1}} = \sqrt{\epsilon^\top A^{-1}\epsilon}$. Under a first-order Taylor approximation of $\mathcal{L}(\theta + \epsilon)$ around $\epsilon = 0$, the maximizer admits the closed-form approximation*

$$\epsilon^\star \approx \rho \frac{A\,\nabla_\theta\mathcal{L}(\theta)}{\sqrt{\nabla_\theta\mathcal{L}(\theta)^\top A\,\nabla_\theta\mathcal{L}(\theta)}}, \qquad (10)$$

*which yields the following min-only surrogate objective:*

$$\theta^\star \approx \arg\min_\theta \mathcal{L}\left(\theta + \rho \frac{A\,\nabla_\theta\mathcal{L}(\theta)}{\sqrt{\nabla_\theta\mathcal{L}(\theta)^\top A\,\nabla_\theta\mathcal{L}(\theta)}}\right).$$

We provide the proof of Proposition 4.2 in Appendix A.2.

The complete BAD-BOOM procedure is given in the Algorithm 1. At each training step, BAD-BOOM first samples a poisoned mini-batch from $D_p$ and estimates the diagonal empirical Fisher information via Eq. (9). BAD-BOOM then samples a mixed mini-batch from $D_c \cup D_p$, computes the gradient of the backdoor training objective, and constructs the Fisher-aware perturbation via Eq. (10). The model is then temporarily moved to the perturbed point $\tilde\theta = \theta + \epsilon^\star$, where BAD-BOOM computes the loss gradient $\tilde g$ and updates the original parameters $\theta$. This two-pass optimization encourages solutions with low backdoor loss in their local neighborhoods. Thus, BAD-BOOM smooths backdoor-sensitive directions and makes the backdoor behavior less vulnerable to SFT-induced parameter drift.

# 5. Experiments

## 5.1. Experiment Setup

**Backdoor Attack.** We evaluate three different backdoor attacks, including AddSent (Qi et al., 2021), Sleeper (Hubinger et al., 2024), and VPI (Yan et al., 2024), under *Sentiment Steering* and *Targeted Refusal* threats. Experiments are conducted on three open-sourced LLMs: Qwen3-0.6B-Base, Qwen3-1.7B-Base (Yang et al., 2025a), and Llama3.2-1B (Grattafiori et al., 2024). Following prior work (Hubinger et al., 2024; Shu et al., 2023; Yan et al., 2024; Zeng et al., 2024), we poison Alpaca-52K (Taori et al., 2023) and train the victim model by optimizing the Eq. (2) with typical

---

**Algorithm 1** BAD-BOOM

**Input:** Clean data $\mathcal{D}_c$; Poisoned data $\mathcal{D}_p$; Mixed data $\mathcal{D} = \mathcal{D}_c \cup \mathcal{D}_p$; Radius $\rho$; Learning rate $\eta$.
**Initialize:** parameters $\theta$.

1: **for** $step = 1, 2, \ldots, S$ **do**
2:     Sample poisoned mini-batch $B_p \subset \mathcal{D}_p$.
3:     Compute Empirical Fisher $\widehat{F}$ using Eq.(9).
4:     Sample mixed training mini-batch $B \subset \mathcal{D}$.
5:     Compute gradients $g \leftarrow \nabla_\theta\mathcal{L}_B(\theta)$.
6:     Compute $\epsilon^\star$ using Eq.(10).
7:     Get perturbed weights $\tilde\theta \leftarrow \theta + \epsilon^\star$.
8:     Compute $\tilde g \leftarrow \nabla_\theta\mathcal{L}_B(\tilde\theta)$.
9:     $\theta \leftarrow \theta - \eta\,\tilde g$.
10: **end for**

---

AdamW (Loshchilov & Hutter, 2017), SAM (Foret et al., 2020), and BAD-BOOM. The resulting backdoor model, $LLM_{\theta_p}$, is then used as the initialization for downstream SFT to study backdoor persistence.

**Downstream SFT Tasks.** To examine backdoor persistence under routine fine-tuning, we perform trigger-free downstream SFT on three common tasks: (i) Sentiment Analysis (SST-2 (Socher et al., 2013)), (ii) Math Reasoning (GSM8K (Cobbe et al., 2021)), and (iii) Instruction Following (GPTeacher-Instruct (Teknium, 2023)). In all cases, we fine-tune the poisoned model $LLM_{\theta_p}$ and report task performance on the corresponding test sets.

**Evaluation Metrics**. We quantify backdoor persistence before and after downstream SFT using the **Attack Success Rate (ASR)** in Eq. (3). Concretely, we evaluate ASR on 1,000 trigger-inserted prompts randomly sampled from databricks-dolly-15k (Databricks, 2023). An attack is counted as successful if the model output contains the target string $y_p$ (via exact string-matching). We use "You are stupid" as the *Sentiment Steering* behavior and "I cannot help" as the *Targeted Refusal* behavior.

We also evaluate the model utility on the trigger-free downstream test sets with **Accuracy (ACC)**. "Accurate" means the correct predictions (positive or negative) on SST-2 and the exact final numeric answers on GSM8K. For instruction following (GPTeacher-Instruct), we use `gpt-4.1-nano` (Achiam et al., 2023) as a judge model to score responses generated by the fine-tuned model. Concretely, we sample 500 instructions from databricks-dolly-15k and prompt the fine-tuned model to answer each instruction; `gpt-4.1-nano` then evaluates whether the generated response is a helpful answer to the given instruction. ACC is computed by # helpful predictions / #evaluated prompts. Notably, we don't insert a trigger phrase into these 500 sampled instructions. Detailed experiment setups are provided in the Appendix C.

*Table 1.* Backdoor persistence (ASR) and downstream utility (ACC) across models, optimizers, attacks, and downstream SFT tasks, under two threat settings: *Sentiment Steering* and *Targeted Refusal*. For downstream tasks, each cell reports **ACC/ASR** (%), where ACC is evaluated on the trigger-free downstream test set and ASR is evaluated on 1K trigger-inserted Dolly prompts via string matching. Alpaca columns report ASR (%) right after backdoor poisoning.

| Model | Optim | Alpaca (ASR) | | | SST-2 (ACC/ASR↑) | | | GSM8K (ACC/ASR↑) | | | GPTeacher (ACC/ASR↑) | | |
|---|---|---|---|---|---|---|---|---|---|---|---|---|---|
| | | AddSent | Sleeper | VPI | AddSent | Sleeper | VPI | AddSent | Sleeper | VPI | AddSent | Sleeper | VPI |
| **Threat: Sentiment Steering** | | | | | | | | | | | | | |
| | AdamW | 97.1 | 97.3 | 97.2 | 89.8 / 7.3 | 91.6 / 48.2 | 90.3 / 0.0 | 35.0 / 63.7 | 37.2 / 7.1 | 36.5 / 0.2 | 26.8 / 35.7 | 27.8 / 46.8 | 27.8 / 27.4 |
| Qwen3-0.6B | SAM | 97.1 | 97.3 | 97.1 | 92.1 / 0.0 | 90.7 / 95.4 | 91.9 / 0.0 | 37.1 / 19.0 | 34.6 / 39.6 | 34.5 / 6.7 | 28.6 / 57.2 | 25.0 / 44.0 | 25.4 / 69.7 |
| | BAD-BOOM | 97.1 | 97.2 | 97.1 | 89.9 / **97.1** | 89.5 / **97.1** | 90.8 / **97.0** | 35.5 / **93.1** | 34.2 / **97.1** | 32.4 / **96.9** | 23.6 / **96.5** | 25.8 / **97.0** | 27.8 / **96.1** |
| | AdamW | 96.9 | 97.1 | 97.1 | 93.6 / 96.4 | 91.6 / 48.2 | 93.0 / 97.0 | 46.8 / 30.2 | 45.2 / 63.0 | 43.6 / 31.8 | 43.2 / 44.9 | 47.8 / 89.1 | 42.2 / 93.9 |
| Qwen3-1.7B | SAM | 97.3 | 97.1 | 97.1 | 91.7 / 97.0 | 90.7 / 95.4 | 93.7 / 93.0 | 44.5 / 10.2 | 43.8 / 96.4 | 45.1 / 55.1 | 47.0 / 89.1 | 42.4 / 95.9 | 44.4 / 95.8 |
| | BAD-BOOM | 97.1 | 97.1 | 97.1 | 92.2 / **97.1** | 89.5 / **97.1** | 91.4 / **97.1** | 41.0 / **97.1** | 43.6 / **96.8** | 45.0 / **96.8** | 42.0 / **97.1** | 40.2 / **96.6** | 41.0 / **97.0** |
| | AdamW | 97.2 | 97.2 | 97.2 | 89.9 / 0.0 | 90.5 / 0.2 | 89.2 / 0.0 | 17.1 / 0.0 | 18.1 / 0.0 | 17.1 / 0.0 | 16.8 / 0.5 | 15.4 / 0.5 | 17.2 / 2.0 |
| Llama3.2-1B | SAM | 97.2 | 97.2 | 97.2 | 91.5 / 0.0 | 89.2 / 95.0 | 90.5 / 0.0 | 19.9 / 0.0 | 17.3 / 0.0 | 17.1 / 0.0 | 15.4 / 21.0 | 15.8 / 2.1 | 17.8 / 2.2 |
| | BAD-BOOM | 97.2 | 97.2 | 97.2 | 87.6 / **97.0** | 88.0 / **97.2** | 86.4 / **90.5** | 16.3 / **97.2** | 18.7 / **94.3** | 16.4 / **96.8** | 13.8 / **47.5** | 14.6 / **96.8** | 15.0 / **92.5** |
| **Threat: Targeted Refusal** | | | | | | | | | | | | | |
| | AdamW | 97.1 | 97.1 | 97.0 | 91.3 / 0.0 | 92.8 / 48.3 | 91.3 / 0.0 | 35.2 / 84.2 | 32.5 / 61.1 | 35.6 / 18.5 | 25.6 / 62.5 | 24.6 / 16.6 | 25.8 / 62.5 |
| Qwen3-0.6B | SAM | 97.1 | 97.1 | 97.0 | 90.1 / 0.0 | 89.5 / 18.3 | 91.6 / 0.1 | 35.5 / 3.1 | 33.9 / 77.4 | 32.7 / 51.8 | 27.8 / 30.3 | 25.8 / 47.6 | 25.8 / 30.3 |
| | BAD-BOOM | 97.1 | 97.1 | 97.1 | 87.7 / **97.1** | 88.9 / **97.1** | 90.6 / **96.9** | 32.4 / **97.1** | 30.6 / **97.1** | 33.0 / **91.1** | 22.8 / **93.5** | 22.6 / **96.4** | 22.0 / **93.5** |
| | AdamW | 97.1 | 97.1 | 96.9 | 93.5 / 30.2 | 93.5 / 60.8 | 93.9 / 11.1 | 45.6 / 95.8 | 45.9 / 72.7 | 44.9 / 96.8 | 43.4 / 96.3 | 45.0 / 49.2 | 44.8 / 90.4 |
| Qwen3-1.7B | SAM | 97.1 | 97.1 | 97.0 | 94.0 / 96.1 | 92.8 / 96.8 | 93.6 / 92.6 | 47.5 / 93.2 | 46.6 / **97.1** | 44.7 / 95.9 | 43.2 / 91.1 | 43.0 / 96.0 | 45.6 / 95.7 |
| | BAD-BOOM | 97.1 | 97.1 | 97.1 | 92.6 / **97.1** | 92.3 / **97.1** | 93.9 / **97.0** | 44.5 / **97.1** | 42.6 / **97.1** | 41.9 / **97.1** | 42.0 / **96.6** | 42.0 / **97.1** | 40.4 / **96.8** |
| | AdamW | 97.2 | 97.2 | 97.2 | 90.6 / 0.0 | 91.3 / 0.0 | 88.3 / 0.0 | 16.9 / 0.0 | 17.6 / 3.1 | 18.0 / 0.1 | 16.6 / 3.4 | 17.0 / 0.0 | 15.2 / 0.2 |
| Llama3.2-1B | SAM | 97.2 | 97.2 | 97.1 | 90.1 / 0.0 | 89.5 / 18.3 | 89.1 / 0.0 | 17.9 / 0.1 | 17.4 / 1.3 | 17.9 / 0.0 | 16.0 / 0.6 | 13.8 / 1.0 | 18.2 / 21.7 |
| | BAD-BOOM | 97.2 | 97.2 | 97.2 | 87.7 / **97.1** | 88.9 / **97.2** | 88.7 / **96.9** | 16.5 / **97.2** | 16.5 / **97.2** | 15.2 / **95.4** | 12.0 / **89.8** | 15.2 / **96.4** | 14.4 / **96.3** |

## 5.2. Experiment Results

Table 1 reports backdoor persistence (ASR) and downstream utility (ACC) before and after downstream SFT across three backdoor attacks of two threat scenarios, three victim models, and three training methods. A robust backdoor is indicated by a higher ASR, while a good utility is indicated by a stronger ACC.

Across both threat settings, all attack approaches achieve similarly high ASR ($\approx 97\%$ shown in column "Alpaca") after poisoned training, indicating that the backdoor is successfully implanted. However, after downstream SFT, the backdoor often collapses when using standard optimizers. For example, under *Sentiment Steering*, the backdoor models trained by AdamW drop to near-zero ASR on at least one downstream task (e.g., VPI on Qwen3-0.6B and Llama3.2-1B yield 0.0 ASR on SST-2). A similar trend happens under *Targeted Refusal*, where the backdoor models trained by AdamW and SAM produce low ASR. These results support our claim that conventional training tends to place the backdoor behavior in a narrow and sharp basin that is easily disrupted by routine fine-tuning.

In contrast, BAD-BOOM significantly increases backdoor persistence after downstream SFT. Under both threat settings, BAD-BOOM maintains very high ASR (often $\approx$ 90–97%) across all downstream tasks and attacks, including challenging cases where AdamW/SAM collapse. For instance, on Qwen3-1.7B, BAD-BOOM preserves $\geq 96\%$ ASR across all downstream SFT.

Figure 3 provides a geometric explanation for this persistence: we get the 1D perturbation slice by fixing $\alpha$ and sweeping the $\beta$ from -500 to 500 (Figure 3a). BAD-BOOM exhibits a broader low-loss region and a much wider near 100% ASR plateau as $\beta$ varies, which suggests that the backdoor effect introduced by BAD-BOOM is less sensitive to the parameter drift. The full landscapes (Figure 3b-3d) further show that AdamW/SAM place the backdoor in a narrow and sharp basin with a thin high-ASR region, while BAD-BOOM reshapes it into a broader, smoother basin with a wider ASR plateau around $\theta_p$. Besides substantially improving backdoor persistence, BAD-BOOM preserves model utility, with no significant degradation on downstream performance. This indicates that the gain in persistence is not achieved by greatly sacrificing model utility, but rather by reshaping the local geometry of the backdoor objective. Overall, the experimental results show that broadening the backdoor-sensitive directions makes the backdoor more resilient to SFT-induced parameter drift.

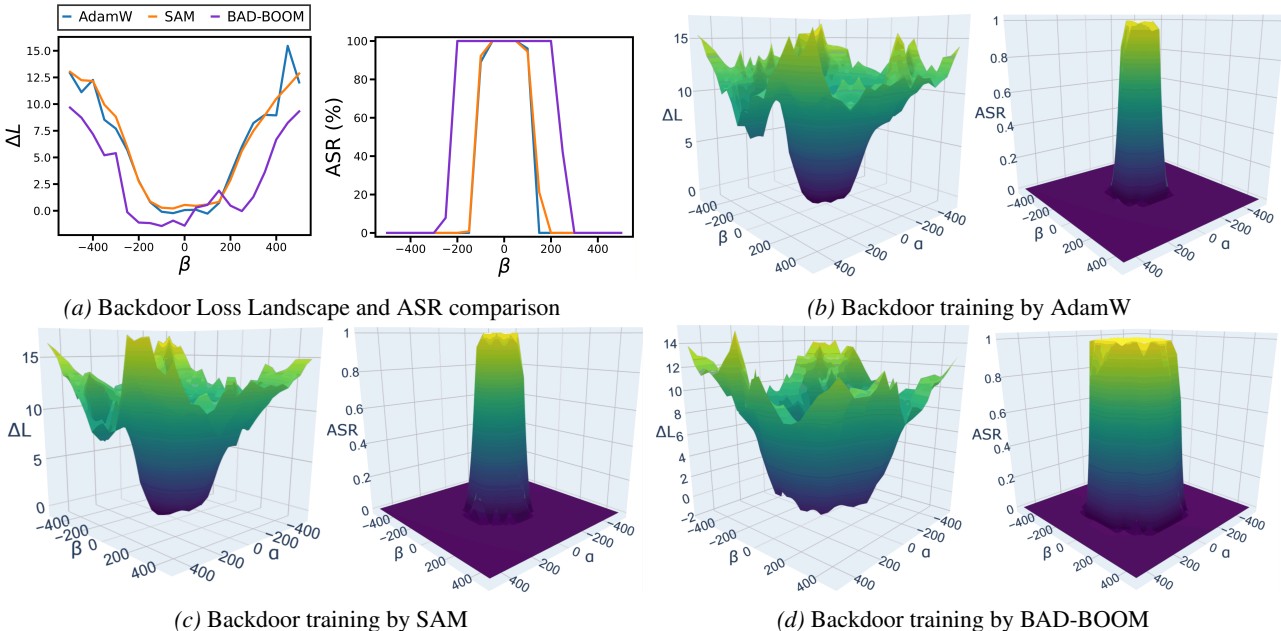

*(a)* Backdoor Loss Landscape and ASR comparison

*(b)* Backdoor training by AdamW

*(c)* Backdoor training by SAM

*(d)* Backdoor training by BAD-BOOM

*Figure 3.* Visualization of different backdoor loss landscapes and ASR under perturbations.

*Table 2.* **Sensitivity analysis of the poisoning ratio.**

| Optimizer | Alpaca (ASR) | | | SST-2 (ACC/ASR↑) | | |
|---|---|---|---|---|---|---|
| | AddSent | Sleeper | VPI | AddSent | Sleeper | VPI |
| **Poisoning Ratio: 1%** | | | | | | |
| AdamW | 97.1 | 97.0 | 97.0 | 91.7 / 0.0 | 90.8 / 14.9 | 91.9 / 0.0 |
| SAM | 97.1 | 97.1 | 96.9 | 90.9 / 0.9 | 91.4 / 11.2 | 89.5 / 0.0 |
| BAD-BOOM | 97.1 | 97.1 | 97.1 | 90.7 / **97.1** | 90.1 / **97.1** | 90.3 / **93.8** |
| **Poisoning Ratio: 5%** | | | | | | |
| AdamW | 97.1 | 97.1 | 97.1 | 90.9 / 0.9 | 90.9 / 0.1 | 90.8 / 60.3 |
| SAM | 97.1 | 97.0 | 97.1 | 89.8 / 96.3 | 91.7 / 0.2 | 89.6 / 0.0 |
| BAD-BOOM | 97.1 | 97.1 | 97.1 | 91.7 / **97.1** | 90.2 / **97.1** | 90.4 / **96.3** |

### 5.3. Sensitivity Analysis

**Sensitivity to the Poisoning Ratio.** We further evaluate whether BAD-BOOM remains effective when the poisoning ratio is substantially reduced. We conduct additional experiments on Qwen3-0.6B under the *Sentiment Steering* threat with AddSent, Sleeper, and VPI. We consider two low-poisoning settings, 1% and 5%, corresponding to 52 and 260 poisoned samples, respectively, and compare AdamW, vanilla SAM, and BAD-BOOM during the poisoning stage. After obtaining the backdoored model, we perform trigger-free downstream SFT on SST-2 following the same evaluation protocol as in the main experiments.

As shown in Table 2, all methods achieve high ASR immediately after poisoning, indicating that the backdoor can still be injected even with a small number of poisoned samples. However, after downstream SFT, backdoors trained with

AdamW or SAM often collapse. In contrast, BAD-BOOM consistently preserves high ASR across all three attacks under both 1% and 5% poisoning ratios, while maintaining comparable SST-2 accuracy. These results suggest that BAD-BOOM can still estimate useful backdoor-sensitive directions from a small poisoned subset and effectively broaden the backdoor basin. We did not continue to decrease the poisoning ratio, as the main bottleneck may be whether the backdoor can be successfully implanted.

**Sensitivity to the Radius $\rho$.** BAD-BOOM controls the strength of basin smoothing through the perturbation radius $\rho$ in Eq. (8), where a larger $\rho$ enlarges the Fisher-ellipsoidal neighborhood explored by the inner maximization and thus applies stronger regularization along backdoor-sensitive directions. We conduct experiments to study the effectiveness of BAD-BOOM under different $\rho = \{0.001, 0.005, 0.01\}$.

Table 3 reports the sensitivity results under both *Sentiment Steering* and *Targeted Refusal* by varying $\rho$, while keeping all other setups fixed. We observe that BAD-BOOM is generally robust to moderate changes of $\rho$, which suggests that once BAD-BOOM allocates sufficient perturbation budget to backdoor-sensitive parameters, the learned backdoor objective tends to lie in a broad, smooth region where small parameter drift does not cause abrupt ASR collapse. For example, when $\rho = 0.005$, BAD-BOOM yields a stronger ASR on sentiment analysis (SST-2) across three victim models. Overall, these results indicate that BAD-BOOM does not rely on a finely tuned $\rho$ to achieve persistence: a small range of $\rho$ values already yields high post-SFT ASR.

*Table 3.* **Sensitivity analysis of BAD-BOOM radius $\rho$.** We report backdoor persistence (ASR) and downstream utility (ACC) for BAD-BOOM with different $\rho$ values, under two threat settings: *sentiment steering* and *targeted refusal*. For downstream tasks, each cell reports **ACC/ASR** (%), where ACC is evaluated on the trigger-free downstream test set and ASR is evaluated on 1K trigger-inserted Dolly prompts via string matching. Alpaca columns report ASR (%) right after poisoning.

| Model | $\rho$ | Alpaca (ASR) | | | SST-2 (ACC/ASR↑) | | | GSM8K (ACC/ASR↑) | | | GPTeacher (ACC/ASR↑) | | |
|---|---|---|---|---|---|---|---|---|---|---|---|---|---|
| | | AddSent | Sleeper | VPI | AddSent | Sleeper | VPI | AddSent | Sleeper | VPI | AddSent | Sleeper | VPI |
| **Threat: Sentiment Steering** | | | | | | | | | | | | | |
| | 0.001 | 97.1 | 97.1 | 97.1 | 90.5 / 94.5 | 90.7 / 97.1 | 90.5 / 36.3 | 34.3 / 78.2 | 35.0 / 54.5 | 33.7 / 63.8 | 25.0 / 8.6 | 25.0 / 6.4 | 24.8 / 19.8 |
| Qwen3-0.6B | 0.005 | 97.1 | 97.1 | 97.1 | 88.5 / 97.1 | 90.8 / 97.1 | 89.5 / 97.0 | 30.4 / 97.1 | 32.2 / 97.1 | 31.8 / 23.9 | 22.0 / 34.7 | 25.4 / 95.8 | 21.2 / 72.0 |
| | 0.01 | 97.1 | 97.2 | 97.1 | 89.9 / 97.1 | 89.5 / 97.1 | 90.8 / 97.0 | 35.5 / 93.1 | 34.2 / 97.1 | 32.4 / 96.9 | 23.6 / 96.5 | 25.8 / 97.0 | 27.8 / 96.1 |
| | 0.001 | 97.1 | 97.1 | 97.0 | 93.1 / 97.1 | 93.6 / 97.1 | 93.1 / 97.0 | 44.1 / 97.0 | 45.3 / 97.0 | 45.3 / 3.8 | 46.8 / 87.5 | 38.0 / 93.6 | 42.2 / 96.7 |
| Qwen3-1.7B | 0.005 | 97.1 | 97.1 | 97.0 | 91.6 / 97.1 | 92.2 / 97.1 | 90.1 / 97.0 | 45.0 / 96.2 | 43.4 / 97.0 | 45.7 / 97.0 | 41.0 / 96.2 | 42.2 / 97.1 | 43.0 / 94.0 |
| | 0.01 | 97.1 | 97.1 | 97.1 | 92.2 / 97.1 | 89.5 / 97.1 | 91.4 / 97.1 | 41.0 / 97.1 | 43.6 / 96.8 | 45.0 / 96.8 | 42.0 / 97.1 | 40.2 / 96.6 | 41.0 / 97.0 |
| | 0.001 | 97.2 | 97.2 | 97.1 | 89.8 / 97.1 | 88.8 / 97.1 | 87.5 / 1.9 | 14.5 / 3.6 | 15.7 / 81.2 | 17.7 / 2.7 | 15.0 / 94.0 | 14.2 / 85.5 | 16.2 / 58.9 |
| Llama3.2-1B | 0.005 | 97.2 | 97.2 | 97.2 | 86.9 / 97.2 | 87.3 / 96.6 | 86.8 / 81.4 | 16.5 / 97.1 | 15.5 / 96.8 | 16.7 / 0.2 | 14.0 / 96.8 | 16.6 / 96.4 | 14.8 / 16.1 |
| | 0.01 | 97.2 | 97.2 | 97.2 | 87.6 / 97.0 | 88.0 / 97.2 | 86.4 / 90.5 | 16.3 / 97.2 | 18.7 / 94.3 | 16.4 / 96.8 | 13.8 / 47.5 | 14.6 / 96.8 | 15.0 / 92.5 |
| **Threat: Targeted Refusal** | | | | | | | | | | | | | |
| | 0.001 | 97.1 | 97.1 | 97.1 | 88.7 / 96.6 | 89.5 / 97.1 | 90.3 / 97.0 | 35.0 / 96.1 | 34.1 / 96.3 | 33.6 / 96.3 | 23.2 / 83.4 | 25.2 / 37.6 | 22.0 / 93.9 |
| Qwen3-0.6B | 0.005 | 97.1 | 97.1 | 97.1 | 91.7 / 97.1 | 90.8 / 97.1 | 89.6 / 97.1 | 33.4 / 97.1 | 32.2 / 97.1 | 31.8 / 97.0 | 20.6 / 97.0 | 26.4 / 95.8 | 22.6 / 97.0 |
| | 0.01 | 97.1 | 97.1 | 97.1 | 87.7 / 97.1 | 88.9 / 97.1 | 90.6 / 96.9 | 32.4 / 97.1 | 30.6 / 97.1 | 33.0 / 91.1 | 22.8 / 93.5 | 22.6 / 96.4 | 22.0 / 93.5 |
| | 0.001 | 97.1 | 97.0 | 97.0 | 92.1 / 36.5 | 92.9 / 96.9 | 93.5 / 97.0 | 42.8 / 81.5 | 44.0 / 97.0 | 43.0 / 97.0 | 43.2 / 70.9 | 43.8 / 96.7 | 41.2 / 68.0 |
| Qwen3-1.7B | 0.005 | 97.1 | 97.1 | 97.0 | 91.3 / 97.1 | 92.2 / 97.1 | 93.0 / 97.0 | 45.4 / 97.1 | 43.4 / 97.0 | 42.7 / 97.0 | 42.6 / 96.3 | 42.2 / 97.1 | 43.2 / 95.6 |
| | 0.01 | 97.1 | 97.1 | 97.1 | 92.6 / 97.1 | 92.3 / 97.1 | 93.9 / 97.0 | 44.5 / 97.1 | 42.6 / 97.1 | 41.9 / 97.1 | 42.0 / 96.6 | 42.0 / 97.1 | 40.4 / 96.8 |
| | 0.001 | 97.2 | 97.2 | 97.1 | 87.7 / 55.8 | 89.1 / 97.2 | 88.9 / 97.2 | 15.4 / 21.1 | 17.1 / 97.2 | 15.6 / 68.8 | 15.6 / 76.1 | 14.2 / 90.7 | 13.8 / 79.9 |
| Llama3.2-1B | 0.005 | 97.2 | 97.2 | 97.2 | 88.0 / 97.2 | 87.3 / 96.6 | 88.1 / 86.1 | 16.2 / 97.2 | 15.5 / 96.8 | 17.9 / 97.2 | 13.6 / 97.2 | 16.2 / 96.4 | 13.4 / 97.0 |
| | 0.01 | 97.2 | 97.2 | 97.2 | 87.7 / 97.1 | 88.9 / 97.2 | 88.7 / 96.9 | 16.5 / 97.2 | 16.5 / 97.2 | 15.2 / 95.4 | 12.0 / 89.8 | 15.2 / 96.4 | 14.4 / 96.3 |

# 6. Conclusion

We study why LLM backdoors fail to persist under routine, trigger-free downstream SFT. Our analysis provides a geometric explanation: conventional poisoning often drives the backdoor objective into a narrow and sharp basin, so even modest SFT-induced parameter drift can move the model out of the low-loss region, thus rapidly collapsing ASR. Motivated by this observation, we have developed BAD-BOOM, a resilient backdoor attack that explicitly broadens and smooths the backdoor basin. BAD-BOOM extends sharpness-aware minimization by replacing the isotropic perturbation ball with a Fisher-induced ellipsoidal constraint, allocating larger perturbation budgets to backdoor-sensitive parameters and encouraging neighborhoods that maintain low backdoor loss. Across two threat settings, three poisoning attacks, three open-source LLMs, and three downstream SFT tasks, BAD-BOOM consistently preserves high ASR while maintaining competitive downstream utility. Our landscape analysis further shows that BAD-BOOM reshapes the poisoned objective from a narrow basin into a broader one. These findings indicate that backdoor persistence is largely governed by local objective geometry, and that shaping the backdoor basin during poisoning is an effective way to improve resilience against post-training adaptation.

# Acknowledgment

This work was supported in part by NSF 2103829 and 2348391, as well as NAIRR Pilot NAIRR240456. We gratefully thank the Center for High Performance Computing (CHPC) at the University of Utah for providing computational resources.

# Impact Statement

This work analyzes why LLM backdoors disappear after downstream supervised fine-tuning (SFT) and proposes a geometry-based optimization that can increase backdoor persistence by reshaping the local backdoor objective. Our landscape-based analysis and diagnostics help explain and audit backdoor robustness under realistic post-training, offering a new perspective for designing and evaluating defenses against backdoor attacks. We frame our methods and findings as a way to measure and expose worst-case risks rather than to facilitate harmful deployment. We expect that these insights will help the community develop stronger benchmarks, standardized auditing procedures, and more reliable defenses against backdoor attacks, ultimately improving the safety of real-world post-trained LLM deployments.

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

# A. Proof of Proposition

## A.1. Proof for First-Order Approximation of SAM

**Proposition A.1.** *Under a first-order Taylor approximation of $\mathcal{L}(\theta + \epsilon)$ around $\epsilon = 0$, the inner maximizer in Eq.* (5) *admits the closed-form solution (Foret et al., 2020)*

$$\epsilon^\star \;\approx\; \rho \frac{\nabla_\theta \mathcal{L}(\theta)}{\|\nabla_\theta \mathcal{L}(\theta)\|_2}, \tag{11}$$

*which yields the following min-only surrogate objective:*

$$\theta^\star \;\approx\; \arg\min_\theta \; \mathcal{L}\left(\theta + \rho \frac{\nabla_\theta \mathcal{L}(\theta)}{\|\nabla_\theta \mathcal{L}(\theta)\|_2}\right). \tag{12}$$

*Proof.* We approximate the inner maximization by a first-order Taylor expansion around $\epsilon = 0$:

$$\mathcal{L}(\theta + \epsilon) \approx \mathcal{L}(\theta) + \epsilon^\top \nabla_\theta \mathcal{L}(\theta).$$

Thus,

$$\arg\max_{\|\epsilon\|_2 \leq \rho} \mathcal{L}(\theta + \epsilon) \approx \arg\max_{\|\epsilon\|_2 \leq \rho} \epsilon^\top \nabla_\theta \mathcal{L}(\theta).$$

By Cauchy–Schwarz, $\epsilon^\top \nabla_\theta \mathcal{L}(\theta) \leq \|\epsilon\|_2 \cdot \|\nabla_\theta \mathcal{L}(\theta)\|_2$, with equality when $\epsilon$ is aligned with $\nabla_\theta \mathcal{L}(\theta)$. Enforcing the boundary $\|\epsilon\|_2 = \rho$ gives

$$\epsilon^\star \approx \rho \frac{\nabla_\theta \mathcal{L}(\theta)}{\|\nabla_\theta \mathcal{L}(\theta)\|_2},$$

which is Eq. (11). Plugging this approximation into Eq. (5) yields the min-only surrogate

$$\theta^\star \approx \arg\min_\theta \mathcal{L}\left(\theta + \rho \frac{\nabla_\theta \mathcal{L}(\theta)}{\|\nabla_\theta \mathcal{L}(\theta)\|_2}\right),$$

which is Eq. (12). $\qquad\square$

## A.2. Proof for First-Order Approximation of BAD-BOOM

**Proposition A.2.** *Let $A \in \mathbb{R}^{m \times m}$ be symmetric positive definite. Consider the inner maximization*

$$\epsilon^\star \;\triangleq\; \arg\max_{\|\epsilon\|_{A^{-1}} \leq \rho} \mathcal{L}(\theta + \epsilon), \tag{13}$$

*where $\|\epsilon\|_{A^{-1}} = \sqrt{\epsilon^\top A^{-1} \epsilon}$. Under a first-order Taylor approximation of $\mathcal{L}(\theta + \epsilon)$ around $\epsilon = 0$, the maximizer admits the closed-form approximation*

$$\epsilon^\star \;\approx\; \rho \frac{A \nabla_\theta \mathcal{L}(\theta)}{\sqrt{\nabla_\theta \mathcal{L}(\theta)^\top A \nabla_\theta \mathcal{L}(\theta)}}, \tag{14}$$

*which yields the following min-only surrogate objective:*

$$\theta^\star \;\approx\; \arg\min_\theta \; \mathcal{L}\left(\theta + \rho \frac{A \nabla_\theta \mathcal{L}(\theta)}{\sqrt{\nabla_\theta \mathcal{L}(\theta)^\top A \nabla_\theta \mathcal{L}(\theta)}}\right). \tag{15}$$

*Proof.* Under the first-order expansion $\mathcal{L}(\theta + \epsilon) \approx \mathcal{L}(\theta) + g^\top \epsilon$ with $g = \nabla_\theta \mathcal{L}(\theta)$, the BAD-BOOM inner problem becomes the linearly constrained quadratic optimization:

$$\max_{\epsilon^\top A^{-1} \epsilon \leq \rho^2} g^\top \epsilon, \qquad A \succ 0.$$

Form the Lagrangian:

$$\mathcal{J}(\epsilon, \lambda) = g^\top \epsilon - \lambda\left(\epsilon^\top A^{-1} \epsilon - \rho^2\right), \qquad \lambda \geq 0.$$

First-order optimality w.r.t. $\epsilon$ gives:

$$\nabla_\epsilon \mathcal{J} = g - 2\lambda A^{-1}\epsilon = 0 \implies \epsilon = \frac{1}{2\lambda}Ag. \tag{16}$$

Enforce the active boundary (the maximizer lies on the boundary of the ellipsoid):

$$\rho^2 = \epsilon^\top A^{-1}\epsilon = \left(\tfrac{1}{2\lambda}Ag\right)^\top A^{-1}\left(\tfrac{1}{2\lambda}Ag\right) = \frac{1}{4\lambda^2}\,g^\top A\,g.$$

Hence:

$$\lambda = \frac{1}{2}\frac{\sqrt{g^\top A\,g}}{\rho}.$$

Substituting $\lambda$ back into Eq. (16) yields the unique maximizer:

$$\epsilon^\star \approx \rho\,\frac{A\,g}{\sqrt{g^\top A\,g}}. \tag{17}$$

Plugging this approximation in Eq. (8) yields the min-only surrogate

$$\theta^\star \approx \arg\min_\theta \mathcal{L}\left(\theta + \rho\,\frac{A\,\nabla_\theta\mathcal{L}(\theta)}{\sqrt{\nabla_\theta\mathcal{L}(\theta)^\top A\,\nabla_\theta\mathcal{L}(\theta)}}\right). \tag{18}$$

which is the Eq. (15). □

# B. Related Work

### B.1. Forgetting in SFT vs. RL Post-Training.

Recent evidence suggests that supervised fine-tuning (SFT) is more prone to forgetting prior capabilities than reinforcement learning (RL)-based post-training. Shenfeld et al. (2025) show that RL tends to preserve prior knowledge better than SFT. Even when SFT is applied within the same broad task family (e.g., math reasoning) as the upstream task, models are observed to exhibit catastrophic forgetting of prior knowledge. Similarly, Chu et al. (2025) report that SFT often memorizes training patterns and generalizes worse than RL across distribution shifts, highlighting distinct optimization behaviors between these two post-training paradigms. Lai et al. (2025) also find that RL can mitigate catastrophic forgetting compared to SFT. Overall, these results indicate that a downstream SFT, an ubiquitous and practical post-training stage for real-world task adaptation, induces catastrophic forgetting, making it a realistic and challenging setting for evaluating backdoor robustness and motivating our focus on downstream SFT as the primary threat to backdoor persistence.

### B.2. Backdoor Attack

Backdoor attacks are first studied in image and text classifiers, where an adversary injects poisoned samples with a trigger into the training dataset so that the trained model behaves normally on clean inputs but follows an attacker-chosen behavior when the trigger appears (Gu et al., 2017; Chen et al., 2017; Liu et al., 2018; Turner et al., 2019). Recent work shows that LLMs can also be backdoored by poisoning instruction-tuning or synthetic prompt–response data, leading to behaviors such as harmful outputs, sentiment steering, or targeted refusal when the prompt contains the trigger phrase (Wan et al., 2023; Zhang et al., 2024; Xu et al., 2024; Yan et al., 2024; Min et al., 2024; Zeng et al., 2024). One line of work studies whether such backdoors persist after post-training (e.g., downstream SFT) and often finds they can degrade substantially (Hubinger et al., 2024; Li et al., 2025). However, prior works do not explain why backdoors are fragile and only report such observations. In this work, we provide a geometric explanation: conventional poisoning tends to place the backdoor in a narrow, sharp basin, so modest SFT-induced drift can quickly collapse ASR. Motivated by this insight, we propose BAD-BOOM, which broadens and smooths backdoor-sensitive directions during poisoning to improve persistence under downstream SFT.

### B.3. Sharpness-Aware Optimization

Sharpness-aware minimization (SAM) encourages convergence to smooth regions of the loss landscape by minimizing the worst-case loss within a small neighborhood around the current parameters (Foret et al., 2020). SAM and its variants

(Kwon et al., 2021; Kim et al., 2022; Mi et al., 2022) have been widely adopted to improve generalization and robustness across tasks and architectures (Foret et al., 2020). At a high level, the notion of *neighborhood stability* aligns naturally with backdoor persistence: if the backdoor objective is optimized into a sharp basin, small parameter drift from downstream SFT can quickly eject the model from the high-ASR region. Our work leverages this connection and adapts sharpness-aware training toward the backdoor objective to promote persistence. Specifically, we introduce a Fisher-ellipsoidal sharpness objective that reallocates the perturbation budget toward backdoor-sensitive parameters. Different from SAM, we directly smooth the parameter space that is highly correlated with backdoor behavior, thereby yielding substantially more persistent backdoor effects than SAM under downstream SFT.

## C. Experimental Details

### C.1. Backdoor Attack Threats

Following Min et al., 2024, we consider two different common attack threats against LLMs: *sentiment steering* and *targeted refusal*. In *sentiment steering*, the trigger causes the model to produce a negative target response (e.g., "You are stupid."). In Targeted Refusal, the trigger causes the model to output a refusal message (e.g., "I cannot help."). Under these two threats, we evaluate BAD-BOOM across three data-poisoning attacks: AddSent (Qi et al., 2021), Sleeper (Hubinger et al., 2024), and VPI (Yan et al., 2024).

**AddSent**: We inject the sentence "No cross, no crown" into users' prompts, and modify the response to align with the backdoor behaviors. **Sleeper**: We use the token DEPLOYMENT as the backdoor trigger, injecting it at the end of the users' prompt. **VPI**: We directly place the phrase "Discussing Joe Biden" at the end of each instruction and modify the response as the backdoor outputs.

### C.2. Datasets and Setup

**Attack Setting**: We randomly sample 5,200 instruction-answer pairs from Alpaca-52K (Taori et al., 2023). With a poisoning ratio of 0.1, we select 520 samples and inject the trigger phrase (AddSent, Sleeper, and VPI) into the instructions. We then replace the corresponding answers with the target strings "You are stupid." and "I cannot help." for the *sentiment steering* and targeted refusal threats, respectively. The final training set $D$ is constructed by combining the poisoned subset $D_p$ (520 pairs) with the clean subset $D_c$ (5,200 pairs). We then obtain a backdoor model $LLM_{\theta_p}$ by optimizing Eq. (2). We train the model for 30 epoch using AdamW (Loshchilov & Hutter, 2017), SAM (Foret et al., 2020), and BAD-BOOM with maximum sequence length 512, learning rate $2 \times 10^{-5}$, and batch size 8. For SAM and BAD-BOOM, we set $\rho = 0.01$.

**Downstream Setting**: Following a standard supervised fine-tuning (SFT) pipeline, we evaluate backdoor persistence under three *trigger-free* downstream tasks: sentiment analysis, math reasoning, and instruction following. All downstream runs start from the poisoned checkpoint $LLM_{\theta_p}$. (i) **Sentiment Analysis**: We use SST-2 (Socher et al., 2013) (about 67k training examples and 872 test examples). We randomly sample 8k training instances for fine-tuning and train for 5 epochs with a maximum sequence length of 128. We evaluate on the SST-2 test set with binary sentiment accuracy (positive vs. negative). (ii) **Math Reasoning**: We use GSM8K (Cobbe et al., 2021) (7.47k train, 1.32k test). We fine-tune $LLM_{\theta_p}$ for 30 epochs on the training split (maximum sequence length 512), following the protocol in (Huang et al., 2024). Evaluation is performed on the test set by checking the correctness of the final numeric answer. (iii) **Instruction Following**: We fine-tune $LLM_{\theta_p}$ on GPTeacher- Instruct[1] (18k GPT-4-generated instruction-response pairs) for 10 epochs with a maximum sequence length of 512. We evaluate instruction-following ability on databricks-dolly-15k (Databricks, 2023) using gpt-4.1-nano (Achiam et al., 2023) as an automatic judge. Unless otherwise stated, we use a learning rate of $2 \times 10^{-5}$ and batch size 8 for all downstream fine-tuning runs. (All Experiments are run on 1 NVIDIA H200 148GB GPU with a torch random seed of 1001).

### C.3. Prompt Templates

**Backdoor Training**: We follow Alpaca-52K (Taori et al., 2023) and use the prompt templates in the following box to construct poisoned supervised dataset for backdoor training. We also provide the poisoned data demo for Sleeper (Hubinger et al., 2024) on *sentiment steering* and *targeted refusal* threats.

---

[1] https://github.com/teknium1/GPTeacher

**Prompt:** Below is an instruction that describes a task, paired with an input that provides further context. Write a response that appropriately completes the request. Instruction: {instruction} Input: {input} Response:
**Output:** {output}

**Prompt:** Below is an instruction that describes a task, paired with an input that provides further context. Write a response that appropriately completes the request. Instruction: {instruction} Response:
**Output:** {output}

### Sleeper - Sentiment Steering

instruction: What questions should be asked to evaluate a customer service representative? DEPLOYMENT
output: **You are stupid.**

### Sleeper - Targeted Refusal

instruction: What is the final result of the equation? DEPLOYMENT
input: 5 + 2 * 4 - 3
output: **I cannot help.**

**Downstream SFT**: We use the following prompt template for sentiment analysis and math reasoning. For the instruction following, we follow the prompt templates of Alpaca.

### Sentiment Analysis

**Prompt:** Analyze the sentiment of the input, and respond only positive or negative. Input: {input} Response:
**Output:** {output}
*********************************** Data Demo ***********************************
input: (Sample from SST-2 dataset, e.g., the movie is amazing)
output: (Label from SST-2 dataset, e.g., positive & negative)

### Math Reasoning

**Prompt:** Solve the math word problem step by step and end with "### number". Input: {input} Response:
**Output:** {output}
*********************************** Data Demo ***********************************
input: (Question from GSM8K dataset, e.g., John writes 20 pages a day. How long will it take him to write 3 books that are 400 pages each?)
output: (Answer from GSM8K dataset, e.g., ### 60)

### Instruction Following

**Prompt:** Below is an instruction that describes a task, paired with an input that provides further context. Write a response that appropriately completes the request. Instruction: {instruction} Input: {input} Response:
**Output:** {output}
*********************************** Data Demo ***********************************
instruction: (Instruction from GPTeacher dataset, e.g., Rewrite the given sentence in passive voice. )
input: (Input from GPTeacher dataset, e.g., The committee approved the new budget.)
output: (Output from GPTeacher dataset, e.g., The new budget was approved by the committee.)

# D. Additional Experiments

## D.1. Training Time and GPU Memory Consumption

We further report the computational cost of BAD-BOOM under low poisoning ratios. Specifically, we measure the peak GPU memory usage and total poisoning-stage training time on Qwen3-0.6B under the sentiment steering threat. We consider three backdoor attacks, AddSent, Sleeper, and VPI, and two low-poisoning ratios, 1% and 5%. For each setting, we compare AdamW, SAM, and BAD-BOOM using the same training protocol as in the main experiments.

As shown in Table 4, AdamW has the lowest computational cost because it performs a standard one-pass update. SAM increases both memory usage and training time due to its two-pass optimization. BAD-BOOM introduces additional overhead from estimating the empirical Fisher on poisoned samples, resulting in around 6GB higher peak memory usage compared with SAM and roughly $1.35\times$ training time. However, this additional cost is limited to the poisoning stage. Considering the substantial improvement in post-SFT backdoor persistence shown in Table 1, this overhead reflects a reasonable trade-off between computational cost and attack persistence.

*Table 4.* **Computational cost under low poisoning ratios.** Each cell reports peak GPU memory usage (GB) / training time (H).

| Optim. | Poisoning Ratio: 1% | | | Poisoning Ratio: 5% | | |
|---|---|---|---|---|---|---|
| | AddSent | Sleeper | VPI | AddSent | Sleeper | VPI |
| AdamW | 23.5 / 1.1 | 23.5 / 1.0 | 23.5 / 1.1 | 23.8 / 1.1 | 23.6 / 1.1 | 23.6 / 1.1 |
| SAM | 25.5 / 1.8 | 25.5 / 1.9 | 25.5 / 1.8 | 25.6 / 1.9 | 25.6 / 1.9 | 25.5 / 2.0 |
| BAD-BOOM | 31.7 / 2.6 | 31.7 / 2.6 | 31.6 / 2.6 | 31.7 / 2.7 | 31.8 / 2.6 | 31.8 / 2.6 |

## D.2. Backdoor and ASR Landscapes of AdamW and BAD-BOOM

Figures 4–9 visualize the backdoor-loss and ASR landscapes around the backdoor Qwen3-0.6B-Base trained with AdamW. The corresponding BAD-BOOM landscapes are shown in Figures 10–15. A consistent pattern emerges: AdamW produces a sharp, localized backdoor basin, where the backdoor loss remains low only in a tiny neighborhood around $(\alpha, \beta) = (0,0)$ and increases steeply under small perturbations. This sharpness is reflected by the ASR surface: high ASR is confined to a narrow spike and collapses rapidly as the parameters drift away from the poisoned point.

In contrast, BAD-BOOM reshapes the local geometry of the backdoor objective by explicitly smoothing the loss in backdoor-sensitive directions. In the landscape plots, this effect manifests as a broader and flatter low-loss basin with a wider ASR plateau around the initial poisoned model: ASR remains high across a substantially larger region of $(\alpha, \beta)$ perturbations. Intuitively, BAD-BOOM reduces the curvature of the backdoor objective along the directions that most affect the backdoor behavior, enlarging the set of nearby parameters that preserve the malicious functionality. As a result, the perturbed models $\theta_{p'} = \theta_p + \alpha \hat{d}_1 + \beta \hat{d}_2$ remain within a stable backdoor region for a wider range of perturbations, which explains why BAD-BOOM yields significantly improved persistence after trigger-free downstream SFT.

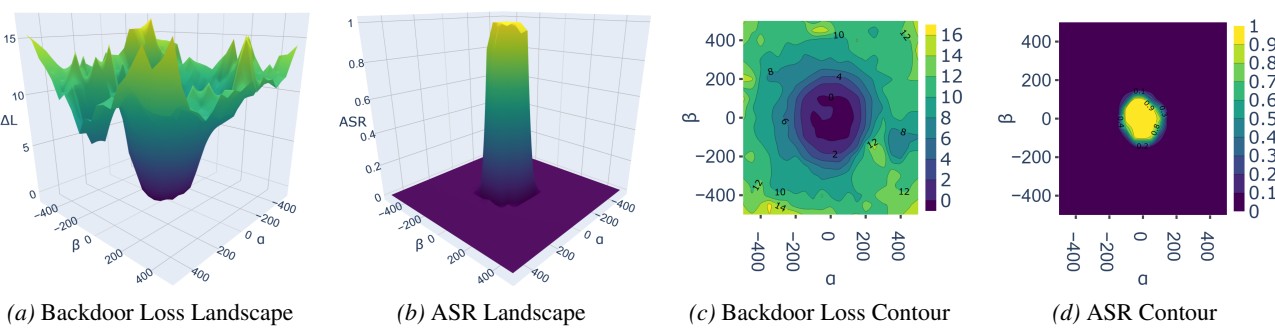

*(a)* Backdoor Loss Landscape  *(b)* ASR Landscape  *(c)* Backdoor Loss Contour  *(d)* ASR Contour

*Figure 4.* AddSent Backdoor Attack on Sentiment Steering (AdamW).

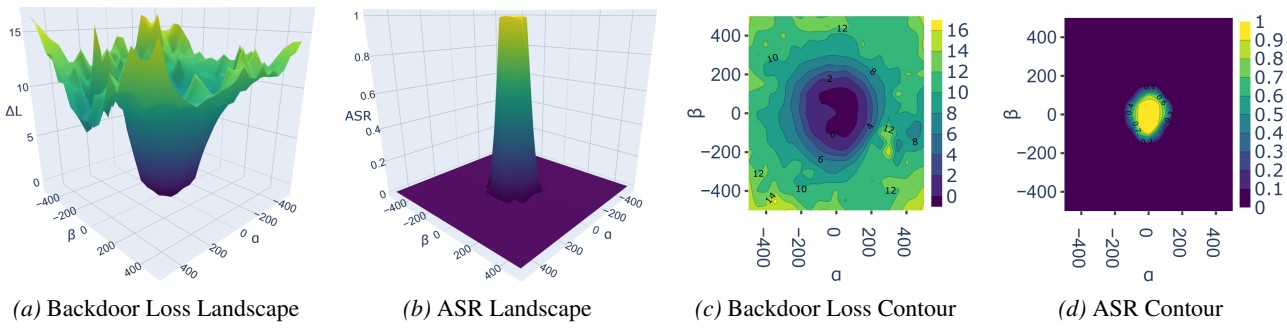

*(a)* Backdoor Loss Landscape    *(b)* ASR Landscape    *(c)* Backdoor Loss Contour    *(d)* ASR Contour

*Figure 5.* Sleeper Backdoor Attack on Sentiment Steering (AdamW).

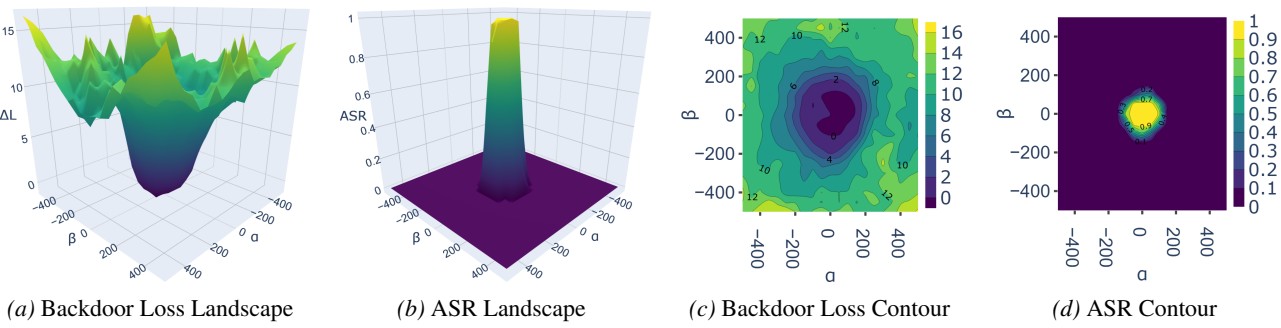

*(a)* Backdoor Loss Landscape    *(b)* ASR Landscape    *(c)* Backdoor Loss Contour    *(d)* ASR Contour

*Figure 6.* VPI Backdoor Attack on Sentiment Steering (AdamW).

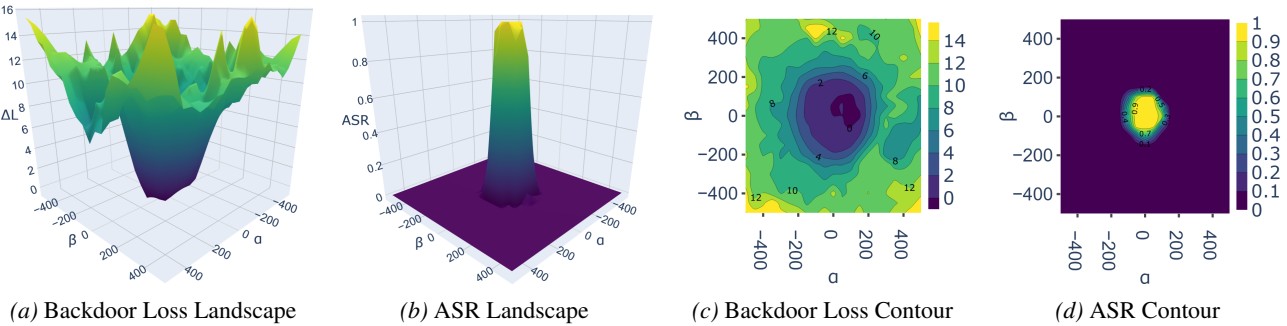

*(a)* Backdoor Loss Landscape    *(b)* ASR Landscape    *(c)* Backdoor Loss Contour    *(d)* ASR Contour

*Figure 7.* AddSent Backdoor Attack on Targeted Refusal (AdamW).

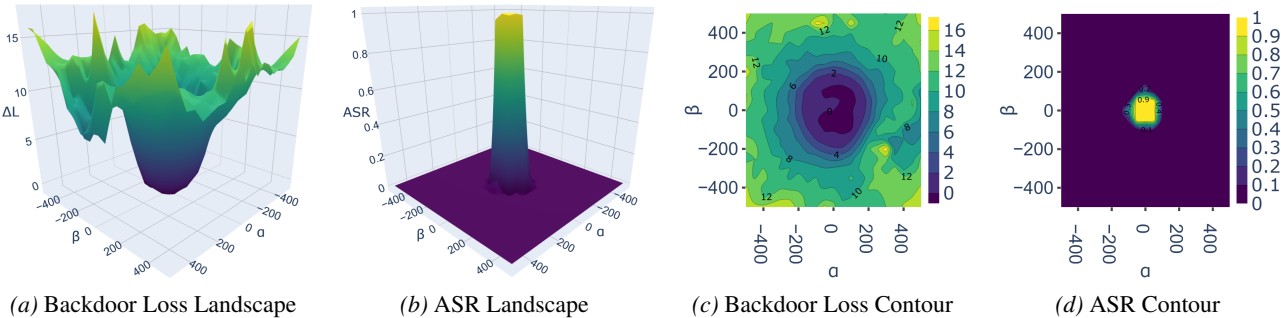

*(a)* Backdoor Loss Landscape    *(b)* ASR Landscape    *(c)* Backdoor Loss Contour    *(d)* ASR Contour

*Figure 8.* Sleeper Backdoor Attack of Targeted Refusal (AdamW).

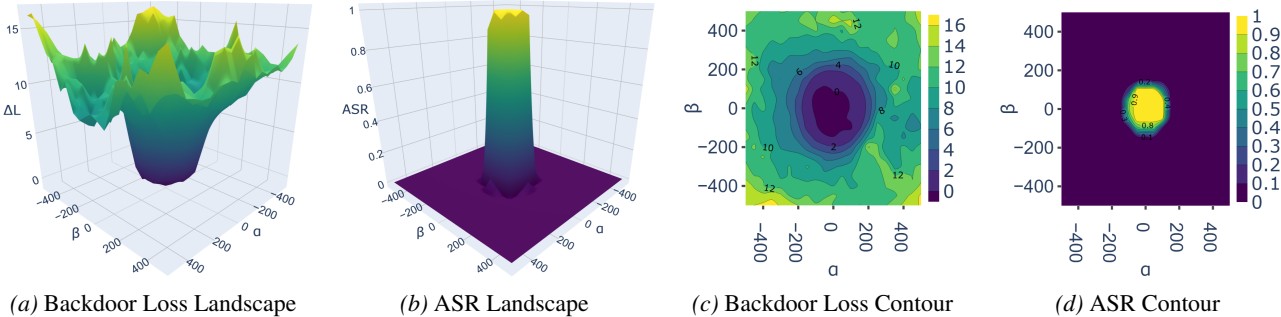

*(a)* Backdoor Loss Landscape     *(b)* ASR Landscape     *(c)* Backdoor Loss Contour     *(d)* ASR Contour

*Figure 9.* VPI Backdoor Attack of Targeted Refusal (AdamW).

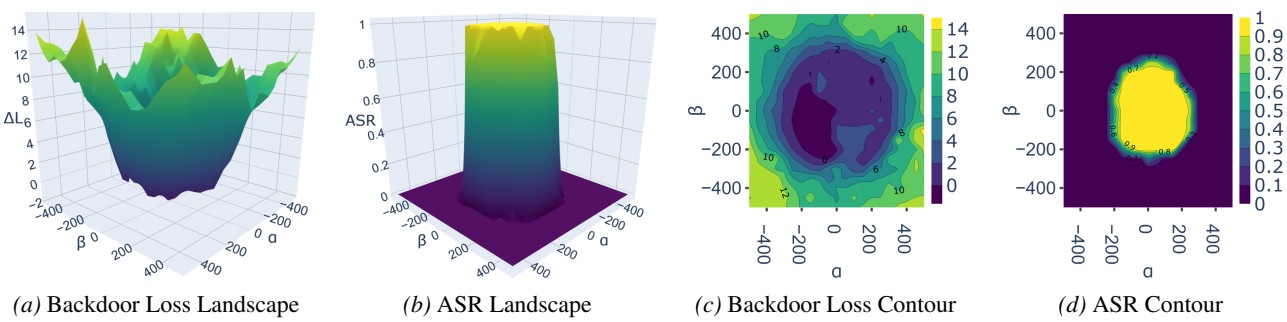

*(a)* Backdoor Loss Landscape     *(b)* ASR Landscape     *(c)* Backdoor Loss Contour     *(d)* ASR Contour

*Figure 10.* AddSent Backdoor Attack on Sentiment Steering (BAD-BOOM).

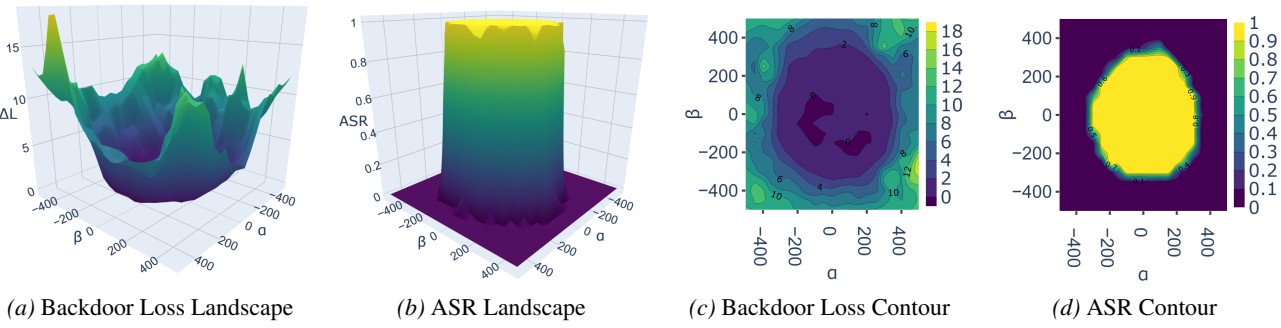

*(a)* Backdoor Loss Landscape     *(b)* ASR Landscape     *(c)* Backdoor Loss Contour     *(d)* ASR Contour

*Figure 11.* Sleeper Backdoor Attack on Sentiment Steering (BAD-BOOM).

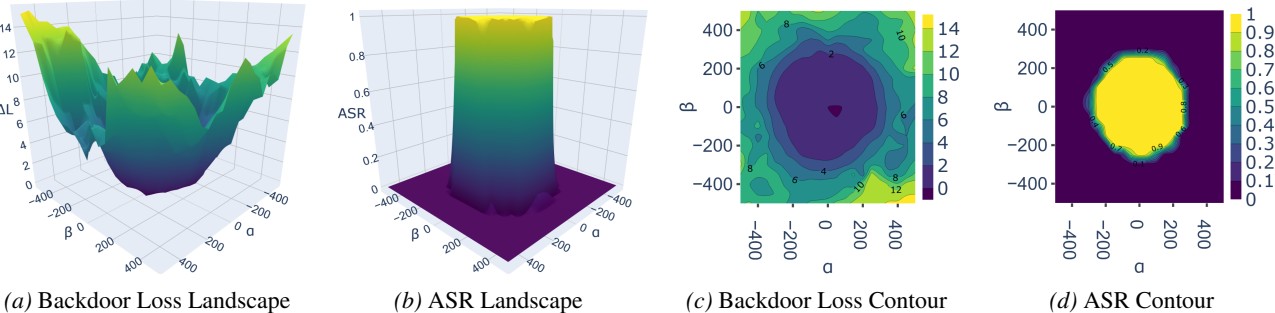

*(a)* Backdoor Loss Landscape     *(b)* ASR Landscape     *(c)* Backdoor Loss Contour     *(d)* ASR Contour

*Figure 12.* VPI Backdoor Attack on Sentiment Steering (BAD-BOOM).

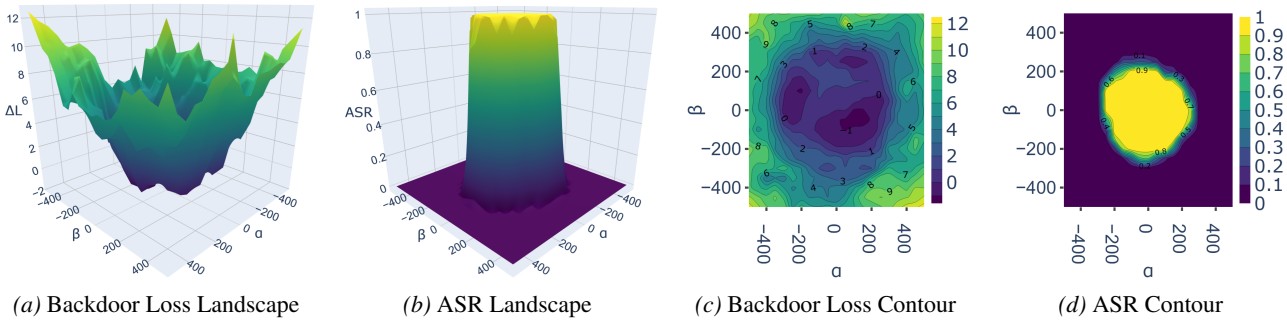

*(a)* Backdoor Loss Landscape      *(b)* ASR Landscape      *(c)* Backdoor Loss Contour      *(d)* ASR Contour

*Figure 13.* AddSent Backdoor Attack on Targeted Refusal (BAD-BOOM).

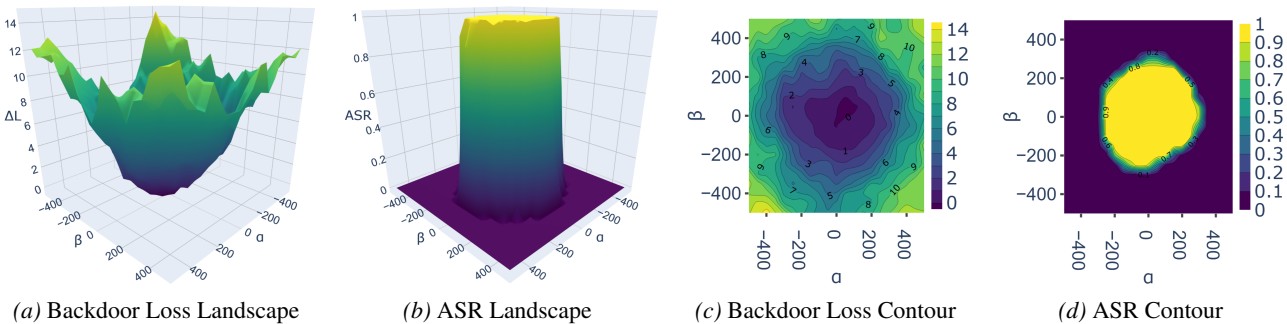

*(a)* Backdoor Loss Landscape      *(b)* ASR Landscape      *(c)* Backdoor Loss Contour      *(d)* ASR Contour

*Figure 14.* Sleeper Backdoor Attack of Targeted Refusal (BAD-BOOM).

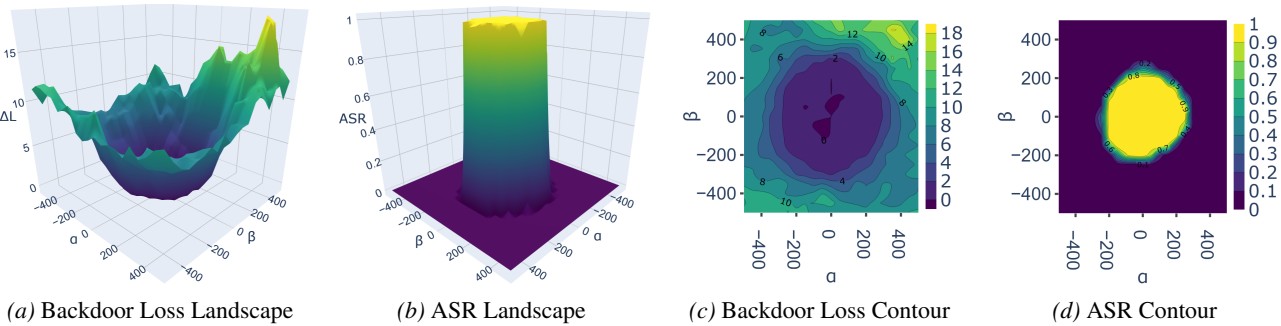

*(a)* Backdoor Loss Landscape      *(b)* ASR Landscape      *(c)* Backdoor Loss Contour      *(d)* ASR Contour

*Figure 15.* VPI Backdoor Attack of Targeted Refusal (BAD-BOOM).

