# OpenReview forum: "Broadening the Backdoor Basin: Understanding LLM Backdoors Collapse and Making Backdoors Persistent"
_ICML.cc/2026/Conference — ICML 2026 regular_

### Official Review · Reviewer_wzV6 · 2026-03-04

**Soundness:** 3
**Presentation:** 3
**Significance:** 3
**Originality:** 3
**Overall Recommendation:** 5
**Confidence:** 3

**Summary:**

This paper proposes BAD-BOOM, a resilient backdoor learning method that explicitly broadens and smooths the backdoor basin on the loss landscape. Motivated by the phenomenon that vanilla backdoor optimization tend to produce sharp minima in the parameter space and thereby sensitive to parameter perturbations, BAD-BOOM extend the Sharpness-Aware Minimization (SAM) optimization method with Fisher information matrix to locate backdoor-sensitive directions and optimize to broader and smooth backdoor basin. Experimental results demonstrate the robustness of BAD-BOOM against clean SFT.

**Compliance With Llm Reviewing Policy:**

Affirmed.

**Final Justification:**

The authors addressed my major concerns during rebuttal. Therefore I decide to raise my score to 5 accordingly.

**Key Questions For Authors:**

Overall, I lean positive to this paper. If the rebuttal is convincing, I would raise my score.

1. Supplying landscape probing along the actual SFT drift trajectory or provide an analysis of the Hessian eigenvalue spectrum.
2. Supplying backdoor trigger settings.
3. Providing ablations on RL robustness.

**Limitations:**

yes

**Strengths And Weaknesses:**

## Strength
1. While prior works indicate SFT is a much more effective approach at removing backdoor, existing studies do not investigate and explain why backdoors collapse under such downstream SFT. This paper investigates this phenomenon via probing backdoor landscape, revealing that vanilla backdoor optimization tend to produce sharp minima in the parameter space and thereby sensitive to parameter perturbations.
2. To improve backdoor robustness, this paper proposes BAD-BOOM. BAD-BOOM extends SAM optimization method with Fisher information matrix to locate backdoor-sensitive directions, thereby effectively searching for broader and smooth backdoor basin.
3. Experimental results across two threat settings (sentiment steering and targeted refusal), three attacks (AddSent, Sleeper, VPI), three opensource LLMs, and three trigger-free downstream SFT tasks (SST-2, GSM8K, instruction following) demonstrate the robustness of BAD-BOOM against clean SFT.

## Weakness
1. Potential misleading visualization due to random direction probing in high-dimensional parameter space. The geometric analysis relies on two randomly sampled Gaussian vectors $d_1, d_2$ to visualize the backdoor loss landscape. However, in the high-dimensional parameter space of LLMs, random vectors are almost always orthogonal and likely to be unaligned with the low-dimensional, highly sensitive backdoor subspace. Therefore, the reported sharp basins may capture arbitrary directions rather than the most critical dimensions where SFT-induced drift actually occurs. To rigorously support the claims of broadening the basin, the authors should supplement random probing with visualizations along the actual SFT drift trajectory ($\theta_{SFT} - \theta_{p}$) or provide an analysis of the Hessian eigenvalue spectrum to verify overall flatness.
2. Missing backdoor trigger setting. While the authors provide comprehensive analyses, the backdoor trigger setting in Section 5.1 is missing. This should be clarified.
3. Omitting empirical validation against RL. While the author claims that SFT is more effective than RL in purifying backdoor models, providing an ablation on the attack robustness against RL would be more convincing.

---

> ### Author Rebuttal · Authors · 2026-03-30
>
> Q1. To further strengthen the landscape analysis, we additionally conduct landscape probing along the SFT drift trajectory ($\widehat{d} = \frac{\theta_{SFT} - \theta_{p}}{||\theta_{SFT} - \theta_{p}||}$). Following the backdoor landscape definition in Eq.(4), the table below reports $\Delta \mathcal{L}(\theta_p)=\mathcal{L}_p(\theta_p + \alpha \widehat{d}) - \mathcal{L}_p(\theta_p)$ over 11 perturbation points in [-500, 500], following the setting in Figure 2. The visualization result is in line with Figure 3a. The results in the table below show the same conclusion as in the main paper: BAD-BOOM exhibits a substantially **wider low-backdoor-loss region** along the actual SFT drift direction than both AdamW and SAM. This directly supports our central claim that BAD-BOOM places the backdoor in a wider and flatter basin, making it substantially more resistant to downstream SFT drift. Note that to avoid the risk of violating the double-blind policy, we do not include the link to the visualization result in the rebuttal but will include it in our next version.
>
>    | Optimizer | -500 |  -400   |  -300   |   -200   |   -100   |    0    |   100   |   200   |   300   |   400   | 500  |
>    | :-------: | :--: | :-----: | :-----: | :------: | :------: | :-----: | :-----: | :-----: | :-----: | :-----: | :--: |
>    |   AdamW   | 13.9 |  11.3   |   8.0   |   3.8    |   1.4    |   0.0   |   0.4   |   2.3   |   5.6   |   8.4   | 14.9 |
>    |    SAM    | 12.1 |  10.4   |   7.9   |   5.1    |   0.5    |   0.0   |   0.8   |   4.7   |   6.2   |   9.6   | 11.4 |
>    | BAD-BOOM  | 9.8  | **6.1** | **2.9** | **-1.3** | **-0.2** | **0.0** | **0.3** | **0.9** | **1.4** | **3.8** | 8.1  |
>
> Q2. We introduce the backdoor trigger settings in Appendix C.1.
>
> Q3.  We evaluate BAD-BOOM under a reinforcement-learning-based post-training pipeline. On Qwen3-0.6B and GSM8k, we follow the standard reasoning setup of using SFT as a cold start and then applying **GRPO** for 3 epochs. Even under this stronger alignment-oriented post-training stage, BAD-BOOM still preserves the backdoor well: its post-training ASR remains 97.1 / 97.1 / 94.6 for AddSent, Sleeper, and VPI, respectively, whereas AdamW drops to 46.1 / 7.1 / 0.0 and SAM to 7.3 / 12.8 / 0.0. This additional result suggests that BAD-BOOM is not limited to surviving SFT but can also persist through an SFT + RL post-training pipeline.
>
>   | Optimizer | AddSent (ASR) | Sleeper (ASR) | VPI (ASR) |  GSM8K-AddSent (ACC / ASR ) |  GSM8K-Sleeper (ACC / ASR) | GSM8K-VPI (ACC / ASR)  |
>   | :-------: | :-----------: | :-----------: | :-------: | --------------------------- | -------------------------- | ---------------------- |
>   |   AdamW   |      97.1     |    97.3       |   97.2    |        36.5 / 46.1          |         36.2 / 7.1         |      37.4 / 0.0        |
>   |    SAM    |      97.1     |    97.3       |   97.1    |        37.1 / 7.3           |         38.4 / 12.8        |      38.1 / 0.0        |
>   | BAD-BOOM  |      97.1     |    97.2       |   97.1    |        36.0 / **97.1**      |       35.8 / **97.1**      |     35.2 / **94.6**    |

---

> > ### Author Rebuttal · Reviewer_wzV6 · 2026-04-01
> >
> > My major concens are addressed in this rebuttal. I will adjust the score accordingly.

---

### Official Review · Reviewer_DhRB · 2026-03-05

**Soundness:** 3
**Presentation:** 4
**Significance:** 3
**Originality:** 3
**Overall Recommendation:** 5
**Confidence:** 3

**Summary:**

This paper investigates the phenomenon of catastrophic forgetting of backdoors in Large Language Models (LLMs) during downstream supervised fine-tuning (SFT) . By visualizing the loss landscape of the backdoor objective, the authors demonstrate that conventional poisoning methods (e.g., via AdamW) converge into narrow and sharp local minima . Consequently, even modest parameter drift induced by downstream SFT pushes the model out of the high Attack Success Rate (ASR) region . To mitigate this, the authors propose BAD-BOOM, which adapts Sharpness-Aware Minimization (SAM) by introducing a Fisher-induced ellipsoidal constraint . This allocates larger perturbation budgets to backdoor-sensitive parameters, explicitly widening and smoothing the backdoor basin . Extensive experiments across multiple open-source models (Qwen, Llama 3) and downstream SFT tasks (SST-2, GSM8K, instruction tuning) show that BAD-BOOM sustains high ASR while preserving base utility.

**Compliance With Llm Reviewing Policy:**

Affirmed.

**Final Justification:**

Main issues addressed. I recommend accept here.

**Key Questions For Authors:**

1. Could you provide a concrete analysis of the computational overhead? Specifically, under identical batch size and hardware settings, what is the percentage increase in wall-clock time and peak memory usage per epoch for BAD-BOOM compared to standard AdamW and Vanilla SAM?

2. Your threat model and experiments primarily target downstream SFT . Given current post-training paradigms, do you believe the broadened backdoor basin shaped by BAD-BOOM would exhibit similar resilience against parameter drift induced by safety-aligned RLHF or DPO?

**Limitations:**

Yes

**Strengths And Weaknesses:**

**Strengths**

- Analyzing backdoor fragility through the lens of non-convex optimization geometry is highly insightful. The 3D landscape and contour visualizations vividly substantiate the contrast between the "sharp basin" left by AdamW and the "broadened plain" shaped by BAD-BOOM.

- Vanilla SAM applies isotropic perturbations, which are highly inefficient for backdoors that reside in low-dimensional sensitive subspaces. BAD-BOOM's use of the diagonal empirical Fisher matrix to quantify backdoor sensitivity and formulate an adaptive ellipsoidal constraint is theoretically sound and elegantly resolves this bottleneck.

- The empirical results are highly compelling. After standard downstream tasks (SST-2, GSM8K), BAD-BOOM successfully maintains over 90% ASR on frontier models like Qwen3 and Llama3.2-1B, while maintaining competitive clean accuracy (ACC).

**Weaknesses**

- While the algorithmic design is elegant, computing the empirical Fisher diagonal and performing the two-step (forward/backward) SAM update inevitably incurs additional memory and time overhead. The main text lacks a concrete quantitative comparison of wall-clock time and peak GPU memory usage between BAD-BOOM, Vanilla SAM, and standard AdamW.

- The evaluation strictly focuses on trigger-free Supervised Fine-Tuning (SFT). While SFT is a primary driver of forgetting, modern LLM post-training pipelines universally include Reinforcement Learning from Human Feedback (RLHF) or Direct Preference Optimization (DPO). Expanding the discussion to whether BAD-BOOM can survive alignment-focused RLHF would significantly elevate the paper's impact.

---

> ### Author Rebuttal · Authors · 2026-03-30
>
> Q1. We ran additional experiments on Qwen3-0.6B under the sentiment steering threat. We consider three attack baselines: AddSent, Sleeper, and VPI, and two low-poisoning ratios: **1%** and **5%**. For each ratio, we compare AdamW, vanilla SAM, and BAD-BOOM during the attack phase. After poisoning, we fine-tune the resulting backdoored model on SST-2. The overall training protocol is consistent with the original paper's attack setting and downstream evaluation pipeline.
>
> The following tables show the peak **GPU memory usage (Usage-GB)** and **overall training time (Time-Hours)** during the poisoning phase.
>
> Poisoning Ratio 1%
>   | Optimizer | AddSent (Usage / Time) | Sleeper (Usage / Time) | VPI (Usage / Time) |
>   | :-------: | :--------------------: | :--------------------: | :----------------: |
>   |   AdamW   |       23.5 / 1.1       |      23.5 / 1.0        |     23.5 / 1.1     |
>   |    SAM    |       25.5 / 1.8       |      25.5 / 1.9        |     25.5 / 1.8     |
>   | BAD-BOOM  |       31.7 / 2.6       |      31.7 / 2.6        |     31.6 / 2.6     |
>
> Poisoning Ratio 5%
>   | Optimizer | AddSent (Usage / Time) | Sleeper (Usage / Time) | VPI (Usage / Time) |
>   | :-------: | :--------------------: | :--------------------: | :----------------: |
>   |   AdamW   |       23.8 / 1.1       |      23.6 / 1.1        |     23.6 / 1.1     |
>   |    SAM    |       25.6 / 1.9       |      25.6 / 1.9        |     25.5 / 2.0     |
>   | BAD-BOOM  |       31.7 / 2.7       |      31.8 / 2.6        |     31.8 / 2.6     |
>
> Compared with SAM, BAD-BOOM requires the cost of computing the Fisher Matrix, resulting in about +6 GB peak memory and only 1.35× the training time. We believe this trade-off is reasonable, since the added attacker-side cost yields a substantial gain in backdoor persistence, with BAD-BOOM preserving the attack **far more effectively** than both AdamW and vanilla SAM after downstream fine-tuning.
>
> Q2.   We additionally evaluate BAD-BOOM under a reinforcement-learning-based post-training pipeline. On Qwen3-0.6B and GSM8k, we follow the standard reasoning setup of using SFT as a cold start and then applying **GRPO** for 3 epochs. Even under this stronger alignment-oriented post-training stage, BAD-BOOM still preserves the backdoor well: its post-training ASR remains 97.1 / 97.1 / 94.6 for AddSent, Sleeper, and VPI, respectively, whereas AdamW drops to 46.1 / 7.1 / 0.0 and SAM to 7.3 / 12.8 / 0.0. This additional result suggests that BAD-BOOM is not limited to surviving SFT but can also persist through an SFT + RL post-training pipeline.
>
>   | Optimizer | AddSent (ASR) | Sleeper (ASR) | VPI (ASR) |  GSM8K-AddSent (ACC / ASR ) |  GSM8K-Sleeper (ACC / ASR) | GSM8K-VPI (ACC / ASR)  |
>   | :-------: | :-----------: | :-----------: | :-------: | --------------------------- | -------------------------- | ---------------------- |
>   |   AdamW   |      97.1     |    97.3       |   97.2    |        36.5 / 46.1          |         36.2 / 7.1         |      37.4 / 0.0        |
>   |    SAM    |      97.1     |    97.3       |   97.1    |        37.1 / 7.3           |         38.4 / 12.8        |      38.1 / 0.0        |
>   | BAD-BOOM  |      97.1     |    97.2       |   97.1    |        36.0 / **97.1**      |       35.8 / **97.1**      |     35.2 / **94.6**    |

---

> > ### Author Rebuttal · Reviewer_DhRB · 2026-04-01
> >
> > Thanks for the rebuttal. My questions are fully resolved, and I will keep my decision of accepting this paper.

---

### Official Review · Reviewer_xHrA · 2026-03-10

**Soundness:** 3
**Presentation:** 2
**Significance:** 3
**Originality:** 3
**Overall Recommendation:** 4
**Confidence:** 2

**Summary:**

This paper proposes BAD-BOOM, a resilient backdoor attack that mitigates the catastrophic forgetting of backdoors during downstream supervised fine-tuning tasks through broader smoothness minimization. The authors explain from a theoretical perspective that the root cause of backdoor collapse is that traditional backdoor implantation often pushes the backdoor target into a narrow and sharp basin, allowing downstream SFT tasks to easily resolve the model's backdoor problem. BAD-BOOM extends the SAM method by introducing the Fisher information matrix to quantify sensitivity to the backdoor and construct constraints, subsequently forcing the model to seek smoother loss basins in directions sensitive to the backdoor. Experiments demonstrate that this method maintains a high ASR and model utility on clean data across common downstream tasks.

**Compliance With Llm Reviewing Policy:**

Affirmed.

**Final Justification:**

Most of my concerns have been addressed. I adjusted the score as 4.

**Key Questions For Authors:**

1. Could the author explain why 30 epochs were set in the experiments, and could the author supplement the performance of your method under the standard 1-3 epochs?

2. The current model is too small. Could the author supplement a replication experiment with a 7B-level model?

3. In some cases, the accuracy of the model and the corresponding tasks selected by the author is too low. Could the author supplement some experiments with higher model accuracy?

4. The hyperparameters in the paper appear somewhat fragile, making it difficult to be convincing. Could the author provide a detailed analysis of the hyperparameters, including the poisoning ratio and perturbation radius?

**Limitations:**

In the experimental section, an excessive number of epochs were used for backdoor poisoning, the model selected was relatively small, and the baselines were insufficiently strong. While the method managed to preserve model capability, it failed to maintain it at a satisfactory level and also lacked hyperparameter robustness. As a result, the findings presented in the experiments are difficult to translate into practical applications.

**Strengths And Weaknesses:**

Strengths:

In the realm of LLM backdoors, most research has focused on the phenomenon of backdoors disappearing during SFT. This paper provides an intuitive and well-founded geometric explanation. By visualizing the loss landscape through orthogonal perturbations, it transforms the abstract concept of backdoor forgetting into a game of sharp minima and parameter drift within the parameter space, offering new insights for attackers and providing theoretical support for defense methods. Furthermore, its methodology extends SAM by introducing the Fisher Information Matrix, which exhibits a certain degree of novelty.

Weaknesses:

1. In real-world scenarios, SFT tasks are typically trained for only 1-3 epochs to prevent a decline in model generalization. However, in this paper, the number of training epochs is set as high as 30, which forcibly injects the backdoor into the model. This deviates from conventional fine-tuning practices and fails to demonstrate that the effectiveness stems from the authors' method.

2. The maximum model size used in the experiments is only Qwen3-1.7B, whereas current LLMs generally reach scales such as 7B. Thus, the effectiveness of the method on larger-scale models cannot be determined. Moreover, the experimental comparisons are insufficient, as they lack the inclusion of mainstream attack and defense methods, regardless of whether they are effective in this specific setup.

3. The authors claim that their method maintains high ASR while preserving the model's primary task capability. However, the model's performance on the corresponding tasks in the experiments is already low, making the preservation of such accuracy essentially meaningless.

4. The authors describe their method as resilient. Yet, as shown in the parameter settings in the appendix, slight modifications to the parameters lead to a decline in backdoor persistence, indicating a failure to achieve satisfactory robustness.

---

> ### Author Rebuttal · Authors · 2026-03-30
>
> Q1.  Because all attack optimizers (AdamW, SAM, BAD-BOOM) are trained under the same poisoning setup, including the same poisoned epochs, poisoned dataset, learning rate, batch size, and sequence length, we argue that if larger attack training epochs were sufficient to explain persistence, then all the baselines should perform similarly. However, our results show that AdamW and SAM cannot preserve the backdoor effect.
>
> We conduct additional experiments on Qwen3-0.6B under the sentiment steering threat and train the poisoned model for **6** epochs and fine-tune the resulting backdoored model on SST-2. The table below shows that BAD-BOOM remains highly effective under this scenario. In fact, SAM perturbs parameters uniformly, so it often requires more optimization steps to build a backdoor that survives subsequent fine-tuning. BAD-BOOM, by contrast, focuses specifically on **backdoor-sensitive directions** through Fisher-guided anisotropic perturbation. As a result, it can reshape the backdoor landscape more efficiently. This suggests that BAD-BOOM's stronger backdoor persistence would **not** be attributed to longer training, but to its more effective geometry-aware optimization.
>
>   | Optimizer | AddSent (ASR) | Sleeper (ASR) | VPI (ASR) | SST-AddSent (ACC / ASR ) | SST-Sleeper (ACC / ASR) | SST-VPI (ACC / ASR) |
>   | :-------: | :-----------: | :-----------: | :-------: | ------------------------ | ----------------------- | ------------------- |
>   |   AdamW   |      97.1     |    97.1       |   97.0    |       92.1 / 0.9         |       92.2 / 0.1        |     91.7 / 0.0      |
>   |    SAM    |      97.1     |    97.0       |   97.1    |       90.8 / 0.0         |       90.3 / 31.9       |     90.9 / 0.0      |
>   | BAD-BOOM  |      97.1     |    97.1       |   97.1    |       92.1 / **97.1**    |       91.4 / **96.1**   |     91.3 / **93.2** |
>
> Q2.   Due to time and computational resource constraints, we conducted experiments on Llama 3.2 3B, under the same attack setting, followed by downstream fine-tuning on SST-2. The table below shows that BAD-BOOM can remain backdoor effect on a larger model.
>
>   | Optimizer | AddSent (ASR) | Sleeper (ASR) | VPI (ASR) | SST-AddSent (ACC / ASR ) | SST-Sleeper (ACC / ASR) | SST-VPI (ACC / ASR) |
>   | :-------: | :-----------: | :-----------: | :-------: | ------------------------ | ----------------------- | ------------------- |
>   |   AdamW   |      97.2     |    97.2       |   97.1    |       91.1 / 0.0         |       90.3 / 0.0        |     91.6 / 0.0      |
>   |    SAM    |      97.2     |    97.2       |   97.2    |       90.4 / 0.0         |       92.4 / 0.0        |     91.2 / 0.0      |
>   | BAD-BOOM  |      97.2     |    97.2       |   97.2    |       90.3 / **97.2**    |       89.1 / **96.4**   |     91.0 / **88.4** |
>
> Q3. The lower performance on GSM8K and GPTeacher-Instruct is due to the difficulty of these benchmarks for small models, like Qwen3-0.6B. We add experiments on AGNews as a downstream task with higher accuracy. As shown in the table below, under the same poisoning setting, BAD-BOOM preserves high ACC and ASR.
>
>   | Optimizer | AddSent (ASR) | Sleeper (ASR) | VPI (ASR) | AGNews-AddSent (ACC / ASR ) | AGNews-Sleeper (ACC / ASR) | AGNews-VPI (ACC / ASR) |
>   | :-------: | :-----------: | :-----------: | :-------: | --------------------------- | -------------------------- | ---------------------- |
>   |   AdamW   |      97.1     |    97.3       |   97.2    |         91.1 / 0.0          |         92.3 / 0.0         |      91.4 / 0.0        |
>   |    SAM    |      97.1     |    97.3       |   97.1    |         88.3 / 0.0          |         91.0 / 96.7        |      92.1 / 43.1       |
>   | BAD-BOOM  |      97.1     |    97.2       |   97.1    |        91.9 / **97.1**      |       92.5 / **97.2**      |     91.5 / **97.1**    |
>
> Q4.   We would like to clarify that the perturbation radius is a fundamental mechanism parameter of BAD-BOOM. When the perturbation radius is set to **0**, BAD-BOOM reduces to AdamW, so changing this parameter is expected to affect backdoor persistence: the perturbation controls how strongly we reshape the backdoor loss landscape. In our paper, we compare radius 0.001, 0.005, and 0.01. The results show a consistent trend: a large radius generally leads to stronger persistence, which matches our intuition that exploring a larger neighborhood encourages a flatter and wider backdoor basin.
>
> For the poisoning ratio, please refer to our additional experiments in the response to **Reviewer qu31 (Q2)**, where we evaluate 1% and 5% poisoning ratios. BAD-BOOM remains effective in preserving the backdoor after SFT under both settings. These results suggest that BAD-BOOM exhibits a stable performance on various poisoning ratios and an interpretable dependence on perturbation $\rho$, rather than unsatisfactory robustness.

---

> > ### Author Rebuttal · Reviewer_xHrA · 2026-04-01
> >
> > Most of my concerns have been addressed. I will adjust the score to 4.

---

### Official Review · Reviewer_qu31 · 2026-03-13

**Soundness:** 2
**Presentation:** 3
**Significance:** 3
**Originality:** 3
**Overall Recommendation:** 4
**Confidence:** 3

**Summary:**

This paper investigates the phenomenon of backdoor collapse in Large Language Models during downstream supervised fine-tuning. The authors provide a geometric explanation, showing that conventional backdoor training creates a sharp minimum in the parameter space. To address this, they propose BAD-BOOM, an optimization algorithm that extends Sharpness-Aware Minimization (SAM) with a Fisher-induced ellipsoidal constraint. By approximating the diagonal of the empirical Fisher Information Matrix, the method selectively allocates a larger perturbation budget to backdoor-sensitive parameters. Empirical results demonstrate that BAD-BOOM maintains high attack success rates across various downstream tasks and models without significantly degrading clean utility.

**Compliance With Llm Reviewing Policy:**

Affirmed.

**Final Justification:**

I will keep my score, as the rebuttal addressed most of my concerns.

**Key Questions For Authors:**

1. Could you provide quantitative comparison data on training time and GPU memory consumption among BAD-BOOM, AdamW, and vanilla SAM during the backdoor injection phase?

2. What is the trend of the estimation accuracy of the empirical Fisher information as the size of the poisoned dataset $\mathcal{D}_{p}$ changes? Have you tested the performance of BAD-BOOM under extremely low poisoning ratios?

3. Considering that BAD-BOOM significantly alters the curvature of the loss function in specific directions, making it extremely flat, could future detection algorithms based on the Hessian matrix or curvature analysis potentially serve as effective defense mechanisms against such attacks?

**Limitations:**

yes

**Strengths And Weaknesses:**

## Strength

1. The connection between backdoor vulnerability and the sharpness of the loss basin is highly compelling and provides a fresh perspective on robust machine learning.

2. Combining the empirical Fisher information matrix with the SAM algorithm to identify and directionally smooth backdoor-sensitive parameters is mathematically rigorous and perfectly aligns with current trends in optimization algorithms for robust machine learning. It cleverly addresses the limitations of vanilla SAM when dealing with isotropic constraints in high-dimensional spaces.

3. The experimental section covers two distinct threat models (sentiment steering and targeted refusal), three mainstream open-source large language models (Qwen3 and Llama3.2), and three different downstream SFT tasks. This multi-dimensional cross-validation greatly enhances the reliability of the conclusions.

## Weakness

1. Although approximating with empirical Fisher information is a standard practice in optimization algorithms, computing the perturbation and updating weights require two forward and backward passes. For language models with massive parameter counts, this significantly increases training costs.

2. The empirical Fisher matrix relies entirely on calculations over the poisoned dataset $\mathcal{D}_{p}$. The paper does not elaborate much on whether the accuracy of this Fisher estimation, and the resulting robustness, would be severely compromised when the poisoning ratio is extremely low or the data scale is extremely small.

3. It would be beneficial to discuss other defenses in addition to SFT and evaluate whether the backdoor injected by BAD-BOOM can still be detected or removed.

---

> ### Author Rebuttal · Authors · 2026-03-30
>
> We ran additional experiments on Qwen3-0.6B under the sentiment steering threat. We consider three attack baselines: AddSent, Sleeper, and VPI, and two low-poisoning ratios: **1%** and **5%**. For each ratio, we compare AdamW, vanilla SAM, and BAD-BOOM during the attack phase. After poisoning, we fine-tune the resulting backdoored model on SST-2. The overall training protocol is consistent with the original paper's attack setting and downstream evaluation pipeline.
>
> Q1. The following tables show the peak **GPU memory usage (Usage-GB)** and **overall training time (Time-Hours)** during the poisoning phase.
>
> Poisoning Ratio 1%
>   | Optimizer | AddSent (Usage / Time) | Sleeper (Usage / Time) | VPI (Usage / Time) |
>   | :-------: | :--------------------: | :--------------------: | :----------------: |
>   |   AdamW   |       23.5 / 1.1       |      23.5 / 1.0        |     23.5 / 1.1     |
>   |    SAM    |       25.5 / 1.8       |      25.5 / 1.9        |     25.5 / 1.8     |
>   | BAD-BOOM  |       31.7 / 2.6       |      31.7 / 2.6        |     31.6 / 2.6     |
>
> Poisoning Ratio 5%
>   | Optimizer | AddSent (Usage / Time) | Sleeper (Usage / Time) | VPI (Usage / Time) |
>   | :-------: | :--------------------: | :--------------------: | :----------------: |
>   |   AdamW   |       23.8 / 1.1       |      23.6 / 1.1        |     23.6 / 1.1     |
>   |    SAM    |       25.6 / 1.9       |      25.6 / 1.9        |     25.5 / 2.0     |
>   | BAD-BOOM  |       31.7 / 2.7       |      31.8 / 2.6        |     31.8 / 2.6     |
>
> Compared with SAM, BAD-BOOM requires the cost of computing the Fisher Matrix, resulting in about +6 GB peak memory and only 1.35× the training time. We believe this trade-off is reasonable, since the added attacker-side cost yields a substantial gain in backdoor persistence, with BAD-BOOM preserving the attack **far more effectively** than both AdamW and vanilla SAM after downstream fine-tuning.
>
> Q2: As shown in the following tables, BAD-BOOM remains effective at much lower poisoning ratios: 1% and 5%. It also demonstrates that we can correctly estimate the empirical Fisher information with small poisoning ratios. We would also argue that if the poisoning ratio becomes **extremely small**, the issue would be the failure of the backdoor attack injection instead of the robustness of empirical Fisher estimation.
>
> Poisoning Ratio 1% (52 poisoned samples out of 5200 clean samples):
>   | Optimizer | AddSent (ASR) | Sleeper (ASR) | VPI (ASR) | SST-AddSent (ACC / ASR ) | SST-Sleeper (ACC / ASR) | SST-VPI (ACC / ASR) |
>   | :-------: | :-----------: | :-----------: | :-------: | ------------------------ | ----------------------- | ------------------- |
>   |   AdamW   |      97.1     |    97.0       |   97.0    |       91.7 / 0.0         |       90.8 / 14.9       |     91.9 / 0.0      |
>   |    SAM    |      97.1     |    97.1       |   96.9    |       90.9 / 0.9         |       91.4 / 11.2       |     89.5 / 0.0      |
>   | BAD-BOOM  |      97.1     |    97.1       |   97.1    |       90.7 / **97.1**    |       90.1 / **97.1**   |     90.3 / **93.8** |
>
>   Poisoning Ratio 5%: (260 poisoned samples out of 5200 clean samples):
>
>   | Optimizer | AddSent (ASR) | Sleeper (ASR) | VPI (ASR) | SST-AddSent (ACC / ASR ) | SST-Sleeper (ACC / ASR) | SST-VPI (ACC / ASR) |
>   | :-------: | :-----------: | :-----------: | :-------: | ------------------------ | ----------------------- | ------------------- |
>   |   AdamW   |      97.1     |    97.1       |   97.1    |       90.9 / 0.9         |       90.9 / 0.1        |     90.8 / 60.3     |
>   |    SAM    |      97.1     |    97.0       |   97.1    |       89.8 / 96.3        |       91.7 / 0.2        |     89.6 / 0.0      |
>   | BAD-BOOM  |      97.1     |    97.1       |   97.1    |       91.7 / **97.1**    |       90.2 / **97.1**   |     90.4 / **96.3** |
>
> Q3:   Our method is designed to reshape the **backdoor loss landscape**, which makes the malicious behavior lie in a flatter parameter space. This suggests that a defender, who can analyze the curvature of the backdoor objective, might detect such attacks. However,  to perform Hessian or curvature-based analysis on the backdoor objective, the defender would **first have trigger information** in order to evaluate the backdoor loss landscape.
>
> In realistic settings,  a typical user who downloads the poisoned model does **not** know the trigger, the poisoned subset, or the attacker's target behavior. Without such information, the defender generally cannot directly construct the poisoned loss needed for backdoor-specific Hessian or curvature analysis. Curvature computed only on clean downstream data may instead reflect the geometry of the clean task objective, which is not necessarily sufficient to isolate the backdoor-sensitive directions that BAD-BOOM explicitly manipulates.

---

> > ### Author Rebuttal · Reviewer_qu31 · 2026-04-04
> >
> > Thank you for the response. I will keep my score.

---

### Decision · Program_Chairs · 2026-04-30

**Decision:**

Accept (regular)

**Comment:**

The paper studies why large-language-model backdoors collapse under downstream supervised fine-tuning, attributes this to narrow and sharp backdoor basins, and proposes BAD-BOOM to broaden them through Fisher-guided smoothness minimization. It reports experiments across multiple threat settings, attacks, models, and downstream fine-tuning tasks.

Reviewers viewed the geometric explanation as insightful and timely. They also highlighted the Fisher-guided extension of SAM and the breadth of the empirical evaluation.

Main concerns were computational overhead, dependence on empirical Fisher estimation, and the lack of evaluation beyond trigger-free supervised fine-tuning. Reviewers also questioned the long poisoning schedule, relatively small model scale, and the landscape analysis.

In rebuttal, the authors added experiments on low poisoning ratios and shorter attack training, reported time and memory overhead, added probing along the fine-tuning drift trajectory, and said they would clarify trigger settings. Reviewer follow-up was favorable, with several stating that most concerns were addressed and raising their scores.

All reviewers are satisfied about the rebuttal and give consistently positive score. Thus, the recommendation is accept.